# Centrioles are frequently amplified in early B cell development but dispensable for humoral immunity

Marina A. Schapfl [1], Gina M. LoMastro[2], Vincent Z. Braun [1], Maretoshi Hirai [3], Michelle S. Levine[2], Eva Kiermaier [4], Verena Labi [1], Andrew J. Holland [2] & Andreas Villunger [1,5] ✉

Centrioles define centrosome structure and function. Deregulation of centriole numbers can cause developmental defects and cancer. The p53 tumor suppressor limits the growth of cells lacking or harboring additional centrosomes and can be engaged by the "mitotic surveillance" or the "PIDDosome pathway", respectively. Here, we show that early B cell progenitors frequently present extra centrioles, ensuing their high proliferative activity and related DNA damage. Extra centrioles are efficiently cleared during B cell maturation. In contrast, centriole loss upon *Polo-like kinase 4 (Plk4)* deletion causes apoptosis and arrests B cell development. This defect can be rescued by co-deletion of *Usp28*, a critical component of the mitotic surveillance pathway, that restores cell survival and maturation. Centriole-deficient mature B cells are proliferation competent and mount a humoral immune response. Our findings imply that progenitor B cells are intolerant to centriole loss but permissive to centriole amplification, a feature potentially facilitating their malignant transformation.

The centrosome acts as the main microtubule-organizing center in animal cells and consists of a mature mother centriole with distal and subdistal appendages and an orthogonally attached daughter centriole, surrounded by proteinaceous pericentriolar material, the PCM[1,2]. In dividing cells, the two centrioles are duplicated once during S-phase to facilitate the formation of a bi-polar mitotic spindle and thus aid the correct segregation of chromosomes. In differentiating cells, they are essential for ciliogenesis and defects in centriole biogenesis are linked to severe developmental abnormalities, including microcephaly and kidney malfunction[3,4]. Aberrations in centriole number frequently occur in cancer and are known to promote chromosomal instability (CIN), provoking aneuploidy[5–7]. Moreover, extra centrosomes have been linked to increased invasiveness and cancer metastasis[8–10]. As such, their number needs to be tightly regulated.

Although cell division can proceed in the absence of centrioles in some circumstances, centrosomes are generally required for sustained proliferation in mammalian cells. Loss of centrioles causes delays in mitosis that activate the mitotic surveillance (*aka* stopwatch) pathway, which promotes p53 stabilization by engaging the p53 binding protein, 53BP1, and the Ubiquitin Specific Peptidase, USP28. In model cell lines, USP28 activity antagonizes MDM2-mediated ubiquitination of p53, leading to its stabilization, transcription of the CDK inhibitor *p21*, and cell cycle arrest in the next G1 phase[11–13]. In vivo, mouse embryos lacking centrioles by loss of SAS4 undergo widespread p53-dependent apoptosis, around day E9.5[14]. Similarly, different mutations leading to centriole biogenesis defects cause apoptosis of neural progenitor cells and microcephaly[15]. In both scenarios, p53-induced apoptosis can be prevented by USP28 co-depletion, ameliorating the related

[1]Institute for Developmental Immunology, Biocenter, Medical University of Innsbruck, Innsbruck, Austria. [2]Department of Molecular Biology and Genetics, Johns Hopkins University School of Medicine, Baltimore, Maryland, USA. [3]Department of Pharmacology, Kansai Medical University, Hirakata, Osaka, Japan. [4]Life and Medical Sciences Institute, Immune and Tumor Biology, University of Bonn, Bonn, Germany. [5]The Research Center for Molecular Medicine (CeMM) of the Austrian Academy of Sciences, Vienna, Austria. ✉e-mail: andreas.villunger@i-med.ac.at

phenotypes[15,16]. How p53 triggers cell death in these settings awaits detailed analysis.

Extra centrosomes, on the other hand, can engage the PIDDosome pathway, e.g., in response to cytokinesis failure or centriole amplification caused by PLK4 overexpression, leading to caspase-2-mediated cleavage of MDM2, p53 stabilization and upregulation of *p21*[17,18]. PID-Dosome activation requires the interaction of PIDD1, its central component, with the transient distal appendage protein, Ankyrin Repeat Domain containing protein 26 (ANKRD26) present at the mature mother[19,20]. PIDD1/ANKRD26 interaction is also key to controlling natural polyploidization of developing and regenerating hepatocytes, defining a clear signaling cascade that connects extra centrosomes to the p53 network[21,22]. Of note, while caspase-2 has been broadly discussed to contribute to p53-induced cell death after DNA damage[23–25], apoptosis initiation in response to extra centrosomes has not been documented.

In hematopoietic cells, centrosomes exert functions that go beyond mitotic spindle pole formation and include erythroblast enucleation[26], control of asymmetric cell division in lymphocytes[27,28], immunological synapse formation[29], as well as cell dendritic cell (DC) migration[30]. Moreover, extra centrosomes have been documented during terminal differentiation of DCs to enhance effective helper T cell activation[30], and in microglia, induced centriole amplification increases their efferocytosis rates[31]. Common to these cell types is the terminally differentiated state that allows them to host extra centrosomes without endangering genome integrity, a situation similar to that found in hepatocytes[32] or osteoclasts[33].

Whether extra centrosomes are limited to terminally differentiated innate immune cells or if they are also found in cells of the adaptive immune system has not been investigated. Hence, we set out to catalog centriole numbers in lymphocytes along different developmental stages in primary and secondary lymphatic organs. B cells develop in the bone marrow, where they progress through defined stages that are clearly separated into phases of proliferation and differentiation[34–36]. The pro B cell and large pre B cell stage mark two early differentiation stages, which are defined by fast proliferation, yet recombination of the immunoglobulin heavy chain (*Igh*) of the immature B cell receptor (pre BCR) requires cell cycle pausing[37]. Expression of the pre-BCR allows cell survival and additional clonal expansion. Subsequently, the cell cycle is stalled, and immunoglobulin light chain recombination is initiated in small, resting pre B cells to complete the assembly of a mature BCR[38]. The successfully rearranged BCR is then expressed as cell surface-bound immunoglobulin (Ig) type M (IgM) on immature B cells that migrate to the spleen to complete initial maturation as follicular or marginal zone B cells. At this point, B cells reach competence to launch humoral immune responses in response to infection via terminal differentiation into Ig-secreting plasmablasts or plasma cells[39].

We noted that proliferating pro B and large pre B cells frequently present with additional centrioles during their ontogeny. However, those are no longer found in more mature developmental stages, beginning from the small resting pre B cell stage onwards.

Apoptosis in proliferating pro and pre B cells with additional centrioles can be blocked by BCL2 overexpression ex vivo, implicating apoptosis in their clearance. Unexpectedly, this cell death does not require p53 or PIDD1, despite pathway competence. Moreover, pre B cells with extra centrioles are never found to mature, even when all apoptosis is blocked in vivo by transgenic BCL2 or MCL1, suggesting additional mechanisms involved in restricting their differentiation in vivo. In contrast, loss of centrioles also promotes BCL2-regulated apoptosis, yet in a strictly p53-dependent manner, arresting early B cell development. Intriguingly, B cell development and humoral immunity can occur in the absence of centrosomes as both can be restored by co-deletion of USP28. Our findings suggest that precise control of centriole counts is a prerequisite for B cell development, but not needed for their effector function in response to antigen-challenge.

## Results

### Early B cell progenitors display imbalances in centriole number

First, we monitored mRNA expression levels of key kinases involved in the centrosome cycle, *Plk1, Plk2,* and *Plk4* (Fig. 1a), and cataloged the number of centrioles in sorted immune cell subsets along lymphocyte ontogeny by immunofluorescence using antibodies against the centriolar marker protein, CP110, and γ-Tubulin to localize centrosomes (Supplementary Fig. S1a). Using this method, more than four centrioles were rarely found in hematopoietic stem cells (HSC), such as LSKs (Lin-Sca1 + c-Kit +) or common lymphoid progenitors (CLP), developing thymocytes, as well as mature T and B cells. In contrast, however, pro and large pre B cells showed frequent centrosome amplification with up to 25 % of cycling large pre B cells displaying > 4 centrioles (Fig. 1b). Of note, this phenomenon was no longer found in resting small pre B cells that exit the cell cycle to rearrange their immunoglobulin light chain locus to express a functional BCR. Signaling via the BCR is needed for maturation and survival, as well as positive and negative B cell selection processes avoiding auto-reactivity. Neither immature IgM+ B cells in the bone marrow nor transitional (T), mature follicular (FO) or marginal zone (MZ) B cells in the spleen, or innate-like B1 B cells in the peritoneum, displayed extra centrioles (Fig. 1b). Taking advantage of double-transgenic mice expressing the FUCCI reporter encoding Geminin-GFP and Cdt1-RFP in all hematopoietic cells[40], we could correlate increased centriole number with the mitotic activity of developing B lymphocyte progenitors where large pre B cells also presented the highest proliferation rate (Fig. 1c and Supplementary Fig. S1b, c). Co-staining of large pre B cells with CEP164, a marker that defines mother centrioles, suggested that the extra centrioles detected were not yet fully mature (Fig. 1d). This finding is indicative of increased centriole nucleation activity that correlated well with higher mRNA levels of *Plk1* and *Plk4* at this developmental stage (Fig. 1a).

### Extra centrioles are linked to proliferation and DNA damage

DNA damage is required for BCR formation[41] and is linked to centrosome amplification[42,43]. To test for possible connections, we co-stained freshly isolated EGFP-CENT1-expressing pro B cells with γH2AX and analyzed them by IF. Interestingly, we found that cells with > 4 centrioles were enriched in the fraction of cells with > 4 γH2AX foci, while in cells with fewer or no γH2AX foci, most cells harbored physiological centriole numbers (2–4) (Fig. 1e). Exploiting the FUCCI reporter system, we also noted that cells with extra centrioles were found primarily in the GFP-positive S/G2/M fraction (Fig. 1f). In addition, using EGFP-CENT1 expressing progenitor B cells, sorted according to fluorescence intensity, confirmed that EGFP-high cells frequently harbored extra centrioles (Supplementary Fig. S1d). These cells were mostly in the G2 fraction of the cell cycle (Supplementary Fig. S1e), as assessed by subtracting phospho-histone H3+ (pH3) cells from the G2/M pool (Supplementary Fig. S1f, g). Furthermore, when pro B cells were sorted based on cell cycle state (G1 vs. S/G2/M) and cultured for 24 h with IL-7, the G1-derived fraction accumulated cells with extra centrioles, while the percentage of cells with 4 centrioles originating from the S/G2/M fraction remained comparable after completion of mitosis (Fig. 1f). This suggests extra centrioles are enriched in S/G2/M cells and can be carried over into the next G1 phase.

Recently, Dwivedi et al. could show[42] that mild replication stress can lead to centrosome amplification in RPE1 cells due to premature centriole disengagement. This effect was abrogated when inhibiting ATR or CHK1 kinase, overriding the G2/M checkpoint. Therefore, we treated apoptosis-refractory BCL2 overexpressing pro B cells with a CHK1 inhibitor for 72h[44] and found that the fraction of cells with extra centrioles was significantly reduced while overall cell cycle distribution remained unchanged (Fig. 1g, h). This suggests that progenitor B cells generate extra centrioles in response to DNA damage, potentially delaying the G2/M transition. In pro B cells, DNA damage can be inflicted physiologically by RAG1/2-activity, introducing double-strand

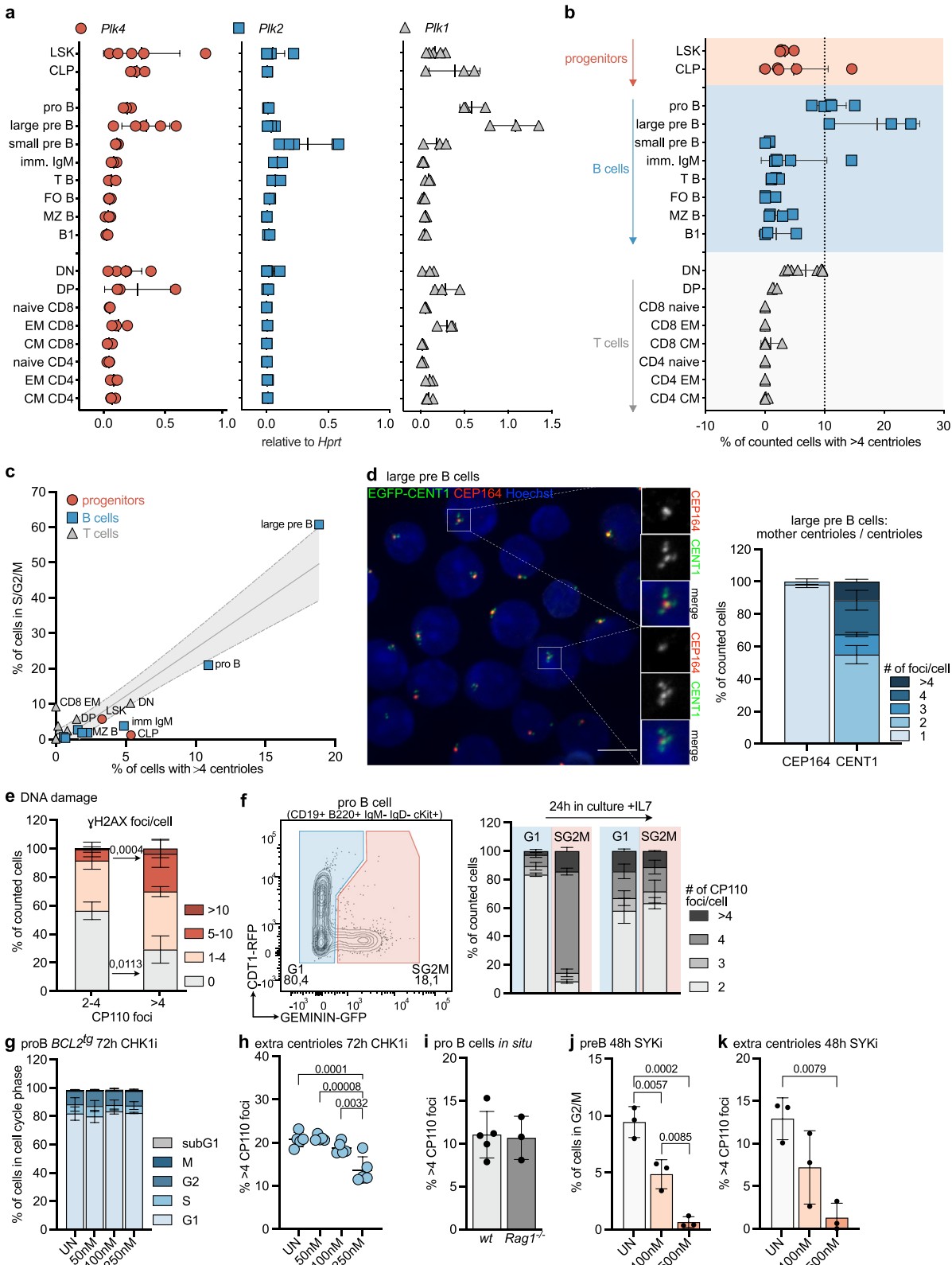

DNA breaks needed for *Igh* rearrangement, or potentially pathological proliferation-associated replication stress. Analysis of RAG1-deficient progenitor B cells indicated that the former was unlikely causal, as *Rag1*⁻/⁻ pro B cells showed an accumulation of extra centrioles comparable to that seen in control cells (Fig. 1i). In contrast, slowing proliferative capacity by using a SYK-inhibitor reduced the percentage of progenitor B cells with > 4 centrioles (Fig. 1j, k). We conclude that extra centrioles most likely arise in response to endogenous DNA damage, caused by the high proliferation rates characterizing early B cell development. While the connection between centriole amplification, DNA damage, and duration of G2 has been noted before in model cell lines[42,43], our observations identify a unique physiological setting where this phenomenon can be observed without the need for genetic manipulation or infliction of genotoxic stress.

**Fig. 1 | B cell progenitors display aberrant centriole numbers. a** qRT-PCR of *Plk4*, *Plk2*, and *Plk1* mRNA normalized to *Hprt* in bone marrow: Lin-Sca1 + c-Kit + (LSK), common lymphoid progenitor (CLP), pro B, large, small pre B and immature IgM + B cells; spleen: transitional (T B) B cells, follicular (FO B) B cells, marginal zone (MZ B) B cells, CD4, CD8 naive, effector memory (EM) and central memory (CM) T cells; thymus: double negative (DN) and double positive (DP) thymocytes; peritoneum: B1 B cells (B1). *n* = 3 (except DN, large pre B (*Plk4* and *Plk2*) *n* = 5 and LSK (*Plk4*, *Plk2*, *Plk1*) *n* = 5) biological replicates **b** Percentage of counted cells with > 4 centrioles. Centriole number was determined by IF staining for γ-Tubulin, CP110 antibody staining, and Hoechst. *n* = 3 (except T B *n* = 4 and DN, imm IgM + , pro B, CLP *n* = 5). **c** Correlation of fraction of cells with > 4 centrioles with a fraction of cells in S/G2/M, determined in mice expressing a FUCCI reporter. *n* = 3. **d** Immunofluorescence quantification of EGFP-CENT1 + large pre B cells stained with αCEP164. Scale bar = 5 μm. **e** Quantification of γH2AX foci in cells with ≤ 4 centrioles or 2–4

centrioles by IF. *n* = 3. **f** Representative dot-plots identifying FUCCI + pro B cells in G1 and S/G2/M. Centriole distribution was assessed directly after sorting and after 24 h in culture by IF using αγ-Tubulin, αCP110, and Hoechst. *n* = 3. **g** Cell cycle profile of pro B cells after 72 h in culture with different concentrations of CHK1i (PF 477736). UN = untreated, *n* = 3. **h** The fraction of cells with > 4 EGFP-CENT1 foci was determined by IF. *n* = 5 (50 vs. 250 nM *p* < 0.0001 = . 000081). **i** Percentage of pro B cells from *wt* and *Rag1*⁻/⁻ animals with > 4 CP110 foci, stained as in (**b**). *n* = 6 (*wt*) *n* = 3 (*Rag1*⁻/⁻). **j** Pre B cells expanded on OP9 cells treated with SYK-inhibitor (R406). The percentage of G2/M cells was analyzed via flow cytometry. *n* = 3. **k** Percentage of SYKi-treated pre B cells with extra centrioles. *n* = 3. Error bars in all panels represent mean ± SD of biological replicates (individual mice) One-way-ANOVA Tukey's multiple comparisons test (**h**, **j**, **k**) and Two-way-ANOVA, Bonferroni's multiple comparisons (**e**). Source data are provided as a Source Data file.

## Extra centrioles are tolerated upon apoptosis inhibition

We rationalized that cycling progenitor B cells with abnormal centriole number may either undergo p21-dependent cell cycle arrest or p53-induced apoptosis, as they were no longer observed at more mature stages of B cell ontogeny (Fig. 1b). Hence, we explored the behavior of early B cells in relation to centrosome number ex vivo in more detail. Towards this end, we isolated pro B cells from mouse bone marrow of different genotypes by cell sorting and expanded them in the presence of IL-7 to prevent their differentiation. Pro B cells from wt, *Pidd1*⁻/⁻, *p53*⁻/⁻, *p21*⁻/⁻, as well as *vav-BCL2* or *vav-Mcl1* transgenic (*BCL2*ᵗᵍ, *Mcl1*ᵗᵍ) mice, expressing anti-apoptotic BCL2 or MCL1 in all blood cells were used[45,46]. Proliferation rates after cytokine stimulation ex vivo were monitored by flow cytometric analyses of pH3 levels. As noted before, BCL2 overexpressing cells showed a lower mitotic index[47] but proliferated at rates comparable to wild-type cells in response to mitogenic stimulation. p53-deficient and *Mcl1* transgenic cells seemed to proliferate slightly better on day 3, but this phenomenon did not persist over time (Supplementary Fig. S2a, c). Analysis of γH2AX as a DNA damage marker revealed a significant accumulation in *BCL2* transgenic cells, but not in the other genotypes (Supplementary Fig. S2b, d). DNA content analysis showed an increase in G2/M cells in the absence of *Pidd1* or *p53* on day 3 that faded over time (Supplementary Fig. S2e, f). Cell death was completely blocked in BCL2 and MCL1 overexpressing cells, while those lacking p53 showed only a transient survival benefit on day 3 (Fig. 2a). Of note, cell death inhibition correlated well with the maintenance of high centriole numbers in BCL2 and MCL1 transgenic progenitor cells (Fig. 2b and Supplementary Fig. S2g). Our findings suggest that loss of *p53* is less effective in preventing the death of cells with extra centrioles, compared to overexpression of antiapoptotic BCL2 or MCL1 (Fig. 2a, b). The cell death observed was clearly apoptotic, as confirmed by TO-PRO3/AnnexinV staining and the use of the pan-caspase inhibitor QVD, which also enabled these cells to persist for a prolonged time ex vivo and retain extra centrioles (Fig. 2c, d). This suggested that apoptosis restricts the survival of cells with extra centrioles in vitro. The fact that loss of *Pidd1* did not protect cells with extra centrioles was surprising but suggested that these did not mature. Consistently, we never found cells with more than one CEP164-positive centriole per cell in any of the genotypes tested, cell death resistant or not (Fig. 2e). Importantly, loss of *Pidd1* as well as BCL2 overexpression both delayed pro B cell death after cytokinesis failure induced by dihydrocytochalsin B (DHCB) treatment, known to cause polyploidy increases and accumulation of mature centrioles (Fig. 2f–h), essential for PIDDosome activation[17]. We conclude that the PIDDosome pathway is functional in pro B cells but not engaged during early B cell development because the extra centrioles noted do not fully mature.

## Extra centrosomes promote progenitor B cell apoptosis

To define extra centrioles as putative initiators of cell death, we tested if overexpression of PLK4, leading to centriole over-duplication, can trigger apoptosis. Therefore, pro B cells were

isolated from the bone marrow of *Plk4*-transgenic mice[48], allowing conditional PLK4 overexpression in response to Doxycycline (Dox), and expanded them in IL-7. After two days, Dox was added to drive excessive centriole biogenesis. *Plk4* mRNA was found to increase on day 5 and day 7, leading to a detectable increase in centriole numbers (Fig. 3a, b and Supplementary Fig. S3a). Assessment of cell cycle profiles, pH3 levels, and DNA damage using γH2AX, however, failed to reveal significant differences when compared to controls (Supplementary Fig. S3b–d). In contrast, cell death rates were found to be increased over time in cells from *Plk4* transgenic mice (Fig. 3c). The fact that the cell cycle profiles did not differ suggested to us that apoptosis induction prevents pathological centriole accumulation and not cell cycle arrest (Supplementary Fig. S3b). Consistent with this idea, we noted that cell death was strongly reduced by BCL2 overexpression, which correlated well with the maintenance of elevated centriole counts (Fig. 3b, c). Inhibition of cell death by transgenic human *BCL2* confirms that PLK4 overexpression induces apoptosis in progenitor B cells.

Based on this finding, we wondered if we might detect extra centrioles at later developmental B cell stages in situ when apoptosis or cell cycle arrest are perturbed. Therefore, different B cell differentiation stages were isolated from the bone marrow of *wt*, *Pidd1*⁻/⁻, *p53*⁻/⁻, *p21*⁻/⁻, *BCL2*ᵗᵍ or *Mcl1*ᵗᵍ mice by cell sorting and analyzed for centriole numbers using CP110 and γ-Tubulin staining to localize and enumerate centrosomes by IF. Remarkably, none of the genotypes analyzed showed centriole counts that exceeded those found in wild-type progenitor B cells (Fig. 3d). Of note, *BCL2* transgenic pro B cells showed a lower percentage of pro B cells with extra centrioles compared to wild-type (Fig. 3d, f), but the total number of cells was comparable to that found in control bone marrow. While the former is in line with the reduced proliferative index of *BCL2* transgenic lymphocytes[47] (Fig. 3e), the latter indicates a relative increase in their abundance due to improved cell survival. Moreover, *Mcl1* transgenic mice showed a clear increase in the absolute number of pro B cells with extra centrioles in the bone marrow (Fig. 3f). Taken together, our findings document that mitochondrial apoptosis restricts the number of progenitor B cells with extra centrioles in the bone marrow of mice. However, their presence must either prevent B cell differentiation or cause cell clearance by alternative cell death mechanisms, as later developmental stages are devoid of cells with extra centrioles, regardless of genotype tested (Fig. 3d). Alternatively, the centrioles themselves may get cleared during the transition into small resting pre B cells.

## Extra centrioles are gradually cleared during but do not perturb differentiation

To model differentiation in vitro, we expanded BM-derived pre B cells with high vs. low centriole count, exploiting EGFP-CENT1 expression and cell sorting to enrich cells with > 4 centrioles (Fig. 3g, h). Cells were expanded on OP9 feeder cells for 7 days and subsequently deprived of

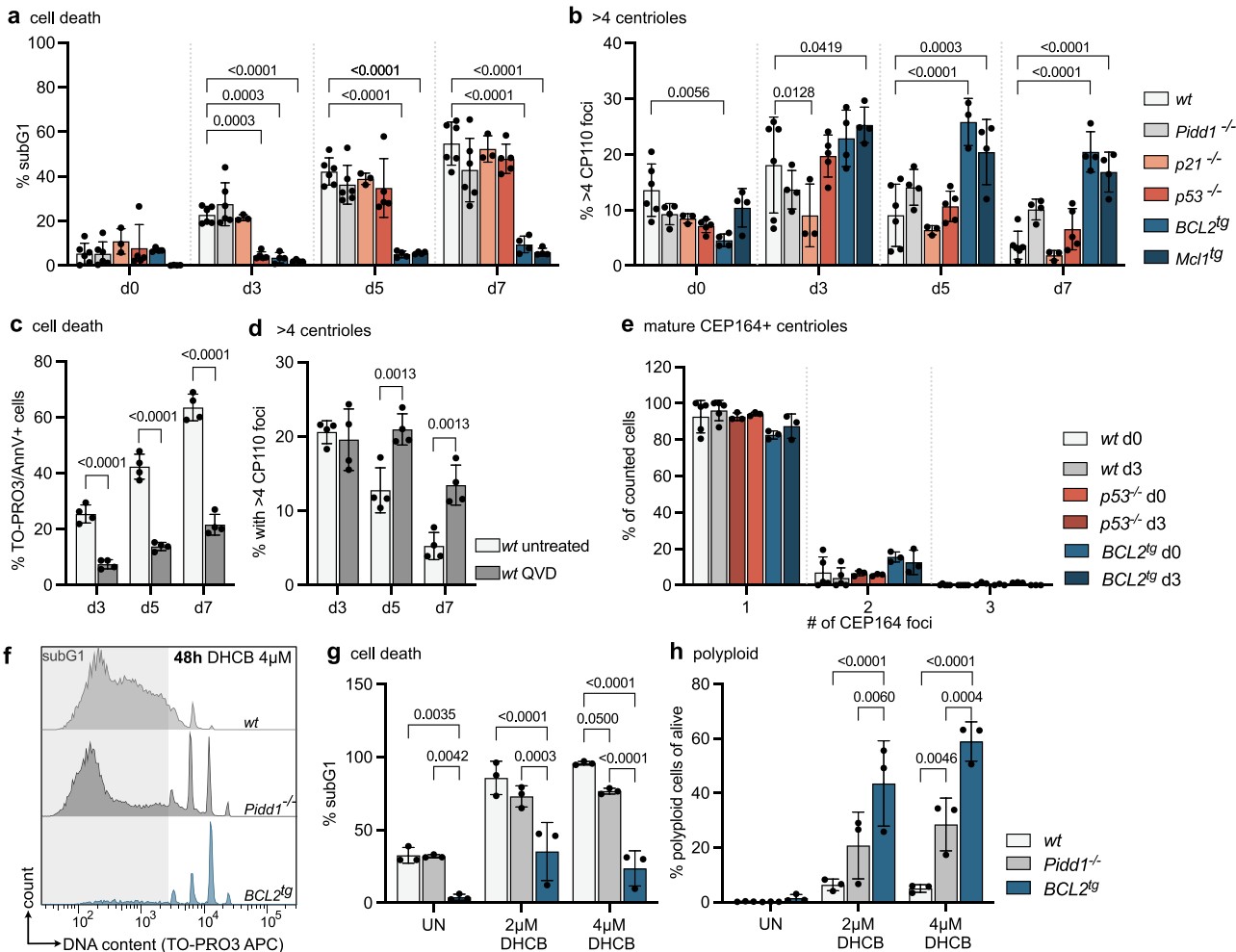

**Fig. 2 | Extra centrosomes correlate with apoptosis induction in progenitor B cells. a** Pro B cells were FACS-sorted from mice of the indicated genotypes, and flow cytometry was used to determine the fraction of dead pro B cells (subG1) after sorting (d0) or in culture after 3, 5, and 7 days (d3, d5, d7) with IL-7. wild type *wt* (*n* = 5), *Pidd1*⁻/⁻ (*n* = 6), *p21*⁻/⁻ (*n* = 3), *pS3*⁻/⁻ (*n* = 5), *BCL2*ᵗᵍ (*n* = 4) and *Mcl1*ᵗᵍ (*n* = 4). **b** Percentage of counted cells with more than 4 centrioles, determined by immunofluorescence with ɣ-Tubulin and CP110 antibody staining and Hoechst nuclear staining. *wt* (*n* = 6), *Pidd1*⁻/⁻ (*n* = 4), *p21*⁻/⁻ (*n* = 3), *pS3*⁻/⁻ (*n* = 5), *BCL2*ᵗᵍ (*n* = 3 d5, *n* = 4 d0, d3, d7) and *Mcl1*ᵗᵍ (*n* = 4). **c** TO-PRO3 + /Annexin V + untreated or QVD-treated *wt* pro B cells were assessed via flow cytometry after 3,5 and 7 days of culture. *n* = 4; **d** Percentage of cells with more than 4 centrioles, determined by immunofluorescence with ɣ-Tubulin and CP110 antibody staining and Hoechst nuclear

staining. *n* = 4; **e** Bar graph showing the percentage of *wt* (*n* = 5), *pS3*⁻/⁻ and *BCL2*ᵗᵍ (*n* = 3) pro B cells with 1,2 or 3 mature CEP164 + centrioles directly after sorting (d0) or after 3 days in culture (d3) with IL-7. CEP164 foci were determined by immunofluorescence with ɣ-Tubulin and CEP164 antibody and Hoechst nuclear stain. **f** Flow cytometry histograms of DNA marker TO-PRO3 of pro B cells expanded for 3 days with IL-7 and then treated with DMSO (UN) or 2, 4 μM DHCB for 48 h. **g** Fraction of cells in subG1. *wt*, *Pidd1*⁻/⁻, *BCL2*ᵗᵍ (*n* = 3). **h** Fraction of polyploid cells alive, *wt*, *Pidd1*⁻/⁻, *BCL2*ᵗᵍ (*n* = 3). Error bars in all panels represent the mean ± SD of biological replicates (individual mice). Genotypes were compared by two-way-ANOVA Tukey's multiple comparisons test (**g**, **h**) and Bonferroni's multiple comparisons test (a-b to compare to *wt*, *a* and **c**, **d**). Source data and exact *p*-values (< 0.0001) are provided as a Source Data file.

---

IL-7 to induce differentiation. Flow cytometric analysis of surface-bound IgM was used to identify the fraction of differentiating cells over time, indicating that the presence of extra centrioles did not interfere with the differentiation capacity of these cells (Fig. 3i). We noted, though, that the percentage of cells with extra centrioles disappeared during this process (Fig. 3h). This suggested that these extra centrioles themselves may become degraded, as IL-7-withdrawal stops proliferation during differentiation. Inhibition of autophagosome/lysosome fusion using chloroquine in *BCL2* transgenic pro B cells led to an increase in LC3-puncta, documenting inhibition of autophagic flux (Fig. 4j), but it did not prevent the disappearance of cells with >4 centrioles upon IL-7 withdrawal (Fig. 4k). Similarly, in cell death refractory BCL2 transgenic pro B cells, the addition of MG132 in conjunction with IL-7 deprivation prevented the decrease in the percentage of cells with >4 centrioles. Unfortunately, MG132 addition also

increased the percentage of cells trapped in G2/M, the cell cycle state where extra centrioles arise (Fig. 3l, m). As such, we were unable to separate the two different events from each other and can currently only exclude autophagy as a regulator of centriole degradation during differentiation.

**Centriole loss promotes p53-driven apoptosis in early B cells**
While extra centrioles appear as a recurrent feature in early B cells, the impact of centriole loss on B cell development has not been assessed. Hence, we explored the response of progenitor B cells to PLK4 inhibition using centrinone[49]. In cancer cell lines, subsequent centriole depletion causes mitotic delays due to problems with proper spindle pole formation[7]. This activates the mitotic surveillance pathway to promote p53-induced cell cycle arrest but can also cause apoptosis in neuronal progenitor cells[15], or cancer cells displaying high TRIM37 E3-

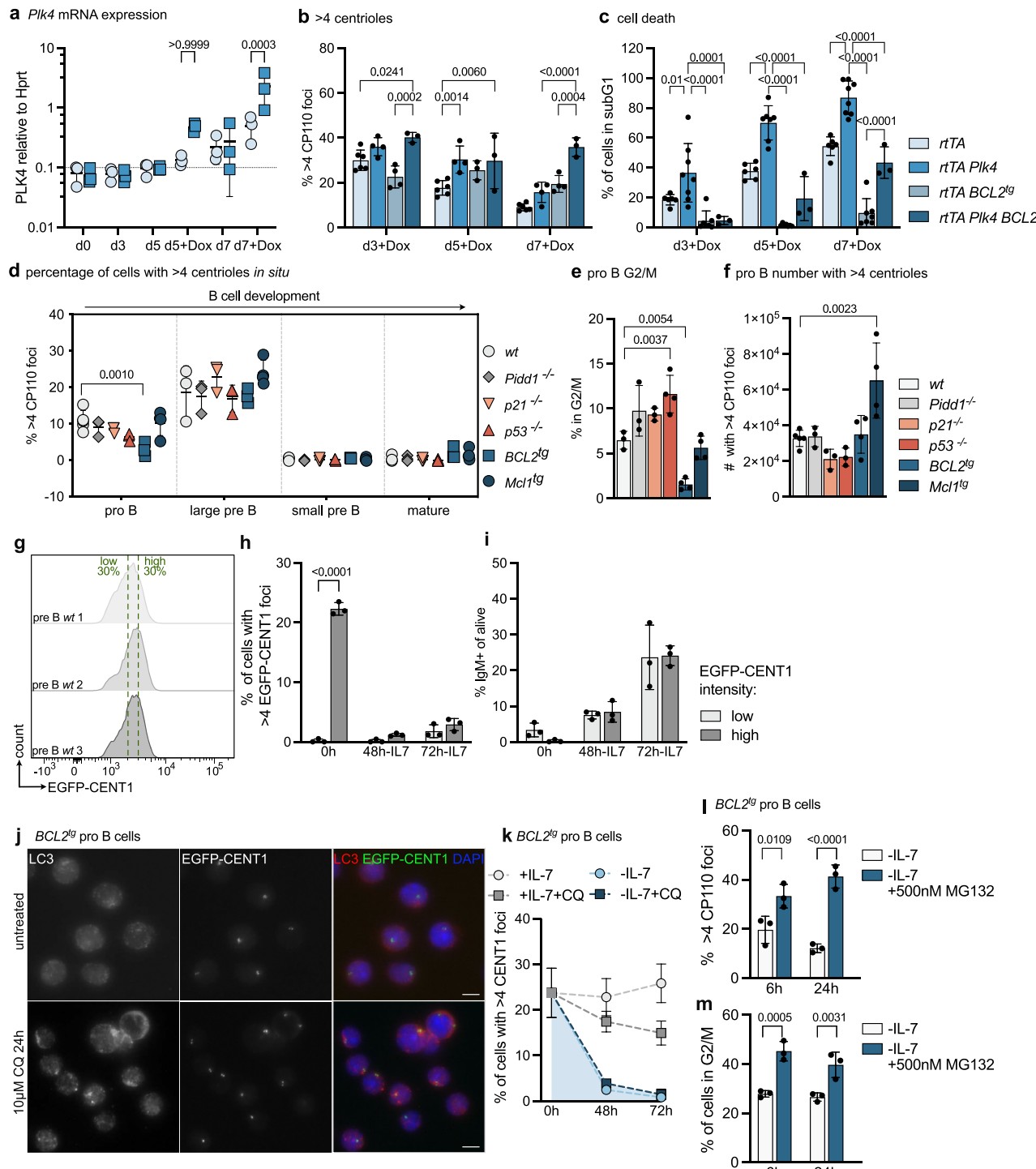

**Fig. 3 | Extra centrosomes promote cell death in progenitor B cells.**
**a** Doxycyclin (Dox) was added after 48 h to pro B culture. qRT-PCR analysis was used to determine *Plk4* mRNA relative to *Hprt*. n = 3. **b** Fraction of counted cells with more than 4 CP110 foci was determined by immunofluorescence (IF). *n = 6 rtTA, n = 4 rtTA Plk4, n = 3 rtTA BCL2^{tg}, n = 3 d5 and n = 4 d3 and d7 rtTA Plk4 BCL2^{tg};* **c** The fraction of dead subG1 cells was determined by flow cytometry. *n = 6 rtTA, n = 8 rtTA Plk4, n = 7 rtTA BCL2^{tg}, n = 3 rtTA Plk4 BCL2^{tg};* **d** FACS-sorted B cells with > 4 centrioles by IF (α-CP110, α-γ-Tubulin). Bone marrow: pro B, large, small pre B; Spleen: follicular B (mature). *n = 3, except wt pro B n = 5 and Mcl1^{tg} (n = 4).* **e** Fraction of pro B in the G2/M phase of the cell cycle after isolation. *n = 3, except n = 4 p53^{-/-}, BCL2^{tg}, Mcl1^{tg};* **f** Calculated number of pro B cells with extra centrioles (fraction of cells with extra centrioles times number of sorted cells). *n = 3 for Pidd1^{-/-}, p21^{-/-}, n = 4 for BCL2^{tg}, Mcl1^{tg} n = 5 for wt;* **g** Flow cytometric histograms of high/low

EGFP-CENT1 pre B cells. **h** Fraction of pre B cells with >4 EGFP-CENT1 + foci in the GFP low vs. high population. **i.** Fraction of pre B cells differentiating to IgM + after 48 or 72 h IL-7 withdrawal by flow cytometry. *n = 3.* **j** Representative IF with α-LC3 and DAPI of *BCL2^{tg}* EGFP-CENT1 + pro B cells untreated (top) or 10 μM Chloroquine treated (bottom) for 24 h. **k** *BCL2^{tg}* pro B cells with more than 4 CENT1 foci. *n = 3.* **l** *BCL2^{tg}* pro B with more than 4 CP110. Cells were expanded for 3 days with IL-7 followed by 6 or 24 h IL-7 withdrawal and treatment with 500 nM MG132.
**m** Fraction of cells in the G2/M phase was determined by flow cytometry. Error bars in all panels represent the mean ± SD of biological replicates (individual mice). Groups were compared by two-way-ANOVA Bonferroni's (**a**, **h−m**) or Tukey's multiple comparisons test (**b−d**) and one-way-ANOVA Tukey's multiple comparisons test (**e**, **f**). Source data and exact *p*-values (< 0.0001) are provided as a Source Data file.

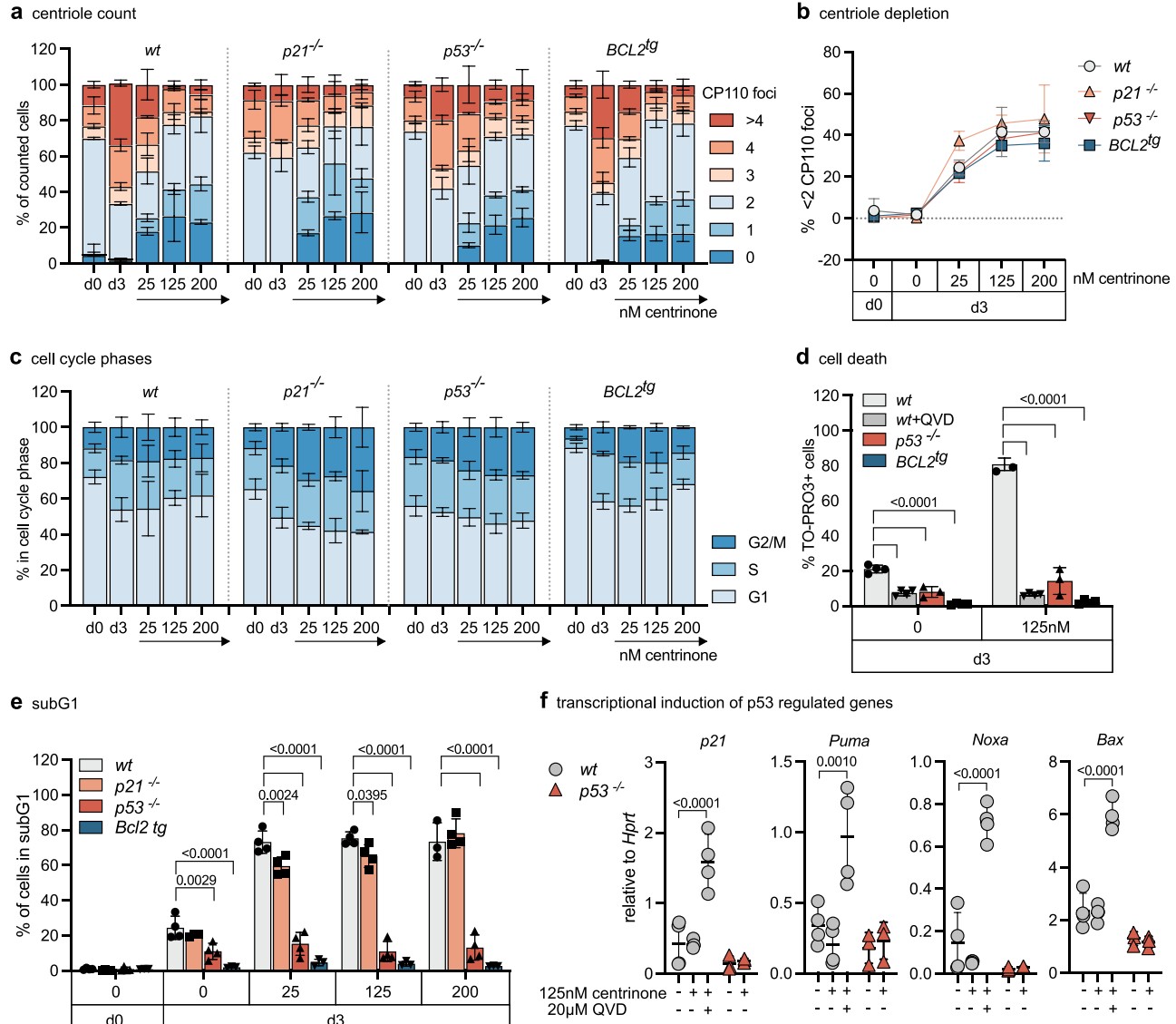

**Fig. 4 | Centriole loss promotes p53-dependent apoptosis in early B cells.**
**a** FACS-sorted pro B cells of *wt*, *p21⁻ᐟ⁻*, *p53⁻ᐟ⁻* and *BCL2ᵗᵍ* were treated with 25, 125 or 200 nM PLK4-inhibitor centrinone for 3 days in presence of IL-7. Centriole distribution was assessed by immunofluorescence with γ-Tubulin, CP110 antibody staining, and Hoechst nuclear staining. *n* = 3 for all genotypes. **b** Fraction of cells with < 2 centrioles (centriole depletion). *n* = 3. **c** Fraction of cells in the G1-, S-, G2/M phase of the cell cycle was determined by flow cytometric cell cycle analysis. *n* = 3 for all genotypes. **d** Bar graph shows the percentage of cells staining positive for the DNA marker TO-PRO3. Pro B cells were cultured for 3 days with IL-7 alone (0) or IL-7 and 125 nM centrinone. *Wt* cells were additionally treated with pan-caspase inhibitor QVD (*wt* + QVD). *n* = 4 (*wt*, *wt* + QVD, *BCL2ᵗᵍ*), *n* = 3 (*p53⁻ᐟ⁻*). **e** Fraction of cells in the subG1 gate was determined by flow cytometric cell cycle analysis. *n* = 4 (*wt*, except 200 nM *n* = 3, *p21⁻ᐟ⁻*, except d3 untreated *n* = 3, *p53⁻ᐟ⁻*), *n* = 3 (*BCL2ᵗᵍ*). **f** qRT-PCR analysis for *p21, Puma, Noxa,* and *Bax* relative to housekeeping gene *Hprt* in pro B cells treated for 48 h with centrinone ± pan-caspase inhibitor QVD. *n* = 4 (*wt*) *n* = 3 (*p53⁻ᐟ⁻*); Error bars in all panels represent mean ± SD of biological replicates (individual mice). Two-way-ANOVA Tukey's multiple comparisons test compared genotypes (**d, e**), but only comparisons to *wt* are shown. Two-way-ANOVA Tukey's multiple comparisons test compared treatments within each genotype (**f**) but only comparisons to untreated control are shown; Source data and exact *p*-values (< 0.0001) are provided as a Source Data file.

ligase activity[50,51]. Hence, we included *p21* and *p53* mutants as well as BCL2 overexpressing cells in our analysis. The addition of centrinone to pro B cell cultures reduced centriole count in a dose and time-dependent manner in all genotypes alike (Fig. 4a, b and Supplementary Fig. S4a). DNA content and pH3 analyses suggested that loss of *p21* or *p53* weakened the mitotic surveillance pathway, as expected (Fig. 4c), leading to an increase in the percentage of cells in mitosis, albeit *p21* loss did not yield statistical significance (Supplementary Fig. S4b, c). BCL2 overexpressing cells were also found to show an increased mitotic index, presumably due to abrogated cell death signaling (Supplementary Fig. S4b, c). Of note, loss of p53 or the inhibition of

caspases using QVD reduced the number of apoptotic cells to a similar degree as BCL2 overexpression (Fig. 4d, e), rendering cells highly resistant to centrinone treatment and also allowing the survival of a small fraction of polyploid cells (Supplementary Fig. S4d). In line, transcriptional induction of p53 target genes *p21, Puma, Noxa*, and *Bax* could be observed in *wt* cells treated with centrinone and QVD, but not in *p53⁻ᐟ⁻* cells. Wild-type cells treated with centrinone in the absence of a caspase inhibitor succumbed rapidly to apoptosis, before p53-dependent transcriptional changes could even be observed (Fig. 4f). Together, this suggests that p53 is the dominant route to apoptosis after centriole depletion.

## Loss of *Plk4* arrests early B cell development

To corroborate these findings in vivo, we analyzed B cell development in mice expressing a floxed *Plk4* allele in combination with the *Mb1Cre* transgene, allowing target gene deletion in the B cell lineage. Analysis of bone marrow and spleen revealed a clear reduction in the overall percentage and number of B cells. Analysis of bone marrow revealed that loss of PLK4 expression led to a general decrease in cellularity and a severe developmental block at the pro/pre B cell stage with a concomitant drop in immature IgM$^+$D$^-$ and IgM$^+$D$^+$ mature recirculating B cells (Fig. 5a–d). A more detailed analysis of progenitor B cell stages documented an accumulation at the pro B cell stage, where cells are highly proliferative and the *Mb1Cre* allele becomes active (Fig. 5c). Within the pre B cell pool, there was a significant drop in small resting and a relative increase in large cycling pre B cells, indicating that these cells may not be able to exit the cell cycle, presumably because they may die off at this stage (Fig. 5d). Corresponding changes were also noted in absolute cell numbers (Supplementary Fig. S5a) and sorted pro B cells lacking *Plk4* did not thrive in the presence of IL-7 ex vivo (Supplementary Fig. S5b). Consistent with a differentiation defect linked to increased cell death, *Mb1CrePlk4F/F* mice showed a significantly reduced B cell percentage and number in the spleen (Fig. 5e, f), with all transitional (T1, T2) and terminal differentiation stages (FO, follicular and MZ, marginal zone B cells) similarly affected (Fig. 5g, h and Supplementary Fig. S5c).

## Centriole-deficient B cells can mediate humoral immunity

Based on our prior analysis, we reasoned that activation of the mitotic surveillance pathway might trigger early progenitor B cell death. Hence, we attempted a genetic rescue experiment by co-deleting *Usp28*. Remarkably, while *Mb1CrePlk4F/F* mice showed a near complete loss of CD19$^+$ B cells in peripheral blood with a parallel increase in CD3$^+$ T cells, co-deletion of *Usp28* restored the percentage of CD19$^+$ cells (Fig. 6a). Similarly, the fraction of pro/pre B cells, cKit$^+$ pro B cells and large pre B cells in the bone marrow, as well as the percentage of B cells in the spleen, were rescued to levels seen in *Plk4F/FUsp28F/F* control mice (Fig. 6b, c). Analysis of steady-state serum immunoglobulin levels revealed that *Mb1CrePlk4F/F* mice still produced near normal levels of IgM, while IgG1 levels were significantly reduced. In *Mb1CrePlk4F/FUsp28F/F* mice, however, both IgM and IgG1 levels were normalized again (Fig. 6c). Hence, we tested whether centriole-depleted B cells were still able to mount an adaptive immune response and immunized animals with the T cell-dependent model antigen, ovalbumin (OVA). Remarkably, while *Mb1CrePlk4F/F* failed to produce OVA-specific immunoglobulins and a concomitant increase in spleen weight, *Mb1CrePlk4F/FUsp28F/F* managed to do so and reached OVA-specific IgG1 levels comparable to those seen in control mice (Fig. 6d).

To confirm that these B cells can survive and mature in the absence of centrioles, we conducted IF staining experiments in CD19$^+$ cells isolated from spleens of *Mb1CrePlkF/F Usp28F/F* mice. Staining for acetylated Tubulin (AcTub), γ-Tubulin, CEP135, or CEP152 all confirmed loss of centrioles in the majority of B cells lacking *Plk4* (Fig. 6e, f and Supplementary Fig. S6A). Moreover, expansion microscopy documented frequent structural alterations in the rare remaining centrioles found in B cells from *Mb1CrePlk F/FUsp28 F/F* mice (Supplementary Fig. S6B).

Finally, we corroborated these findings in a co-culturing system of naïve B cells with CD40 ligand expressing and BAFF-secreting feeder cells, mimicking an induced germinal center (iGC) reaction[52]. This allowed us to determine the dependency of iGC B cells on centrioles for class switch recombination (CSR) and plasmablast generation. Consistent with mouse genetics, centriole loss induced by centrinone treatment did not impact cell cycle distribution (Fig. 6g, h and Supplementary Fig. S7c), nor cell survival (Supplementary Fig. S6c). Moreover, IL-21-driven differentiation into CD138 +

plasmablasts (Supplementary Figs. S6d, e, S7d) or CSR from IgM to IgG1 of GC B cells (Supplementary Fig. S6f–h ) was also unaffected by centriole loss.

To explain the different behavior of progenitor and mature B cells to centriole loss, we reasoned that GC B cells might either express low levels of TRIM37, facilitating acentriolar cell divisions[50,51], or effectively suppress p53 activation in response to centrinone treatment. The latter may be seen as collateral, as BCL6 usually represses p53 transcription to allow for CSR and somatic hypermutation during GC reactions[53]. Consistently with both, GC B cells indeed showed lower *Trim37* mRNA levels and no signs of mounting a transcriptional p53 response. In contrast, pro B cells induced a series of pro-apoptotic p53 target genes (Fig. 6i, j).

Together, this suggests that mature follicular B cells can proliferate, undergo class switch recombination, and differentiate into plasmablasts to secrete immunoglobulins upon antigenic challenge in the absence of centrioles because of a naturally suppressed p53 response, combined with low TRIM37 expression levels.

## Discussion

Here, we provide evidence that B cell development depends on the maintenance of exact centriole counts and that amplified centrioles are a regular feature of cycling pro and large pre B cells. Additional centrioles, however, are no longer detectable at more mature B cell differentiation stages and subsets, starting from small resting pre B cells onwards (Fig. 1). We believe that their clearance is critical to maintain genome integrity as their presence coincides with the differentiation stage where early B cell malignancies frequently arise[54,55]. Consistently, centrosomal abnormalities are a recurrent feature of pre B acute lymphoblastic leukemia (ALL) in humans, and correlate with poor prognosis, but not with a particular genetic makeup of ALL[56,57].

Whether extra centrioles exert a biological function at this developmental stage or if they mark non-functional progenitor cells at risk of failing development or transformation remains unclear. We initially reasoned that such cells may have experienced delays or errors during heavy chain rearrangement, leading to persisting DNA damage, cell cycle delays, subsequent centriole amplification, and cell death. However, our findings using *Rag1* mutant progenitor B cells, showing similar increases in centriole counts (Fig. 1), argue against this hypothesis and rather support the idea that their high proliferation rate may cause DNA damage and delays in G2/M transition. This may allow for premature PLK1-driven centriole disengagement and amplification, as recently reported in the context of mild replication stress[42]. Notably, p53 pathway activity is reported to be epigenetically repressed at this particular differentiation stage by the Polycomb group protein, BMI, to allow *Ig* heavy chain recombination and proliferation of progenitor B cells[58]. This may explain why neither loss of p53 nor PIDD1, which can activate p53 in the context of centrosome amplification[17], provides significant cell death protection, comparable to that seen upon BCL2 or MCL1 protein overexpression (Fig. 2).

Of note, extra centrioles have also been observed in multiple cell types[59]. In multi-ciliated olfactory sensory neurons (OSN) and their immediate neuronal precursors, that can still cycle in their presence, amplification was attributed to high levels of the *SCL/Tal1-interrupting locus* gene (*Stil*) and *Plk4*[60]. *Plk1* and *Plk4* also appear most abundant in cycling progenitor B cells (Fig. 1). Whether the increased mRNA expression levels of centriole biogenesis genes noted are indeed causal for the centriole amplification seen in OSNs or if they solely correlate with the proliferative activity across different progenitor cell types remains to be investigated further. Extra centrioles were not observed in later differentiation stages of B cell development, neither in the absence of PIDD1, p53 nor the presence of transgenic *BCL2* or *Mcl1*, despite a relative increase in the number of these cells at the progenitor B cell stage (Fig. 3). This suggests that inhibition of apoptosis

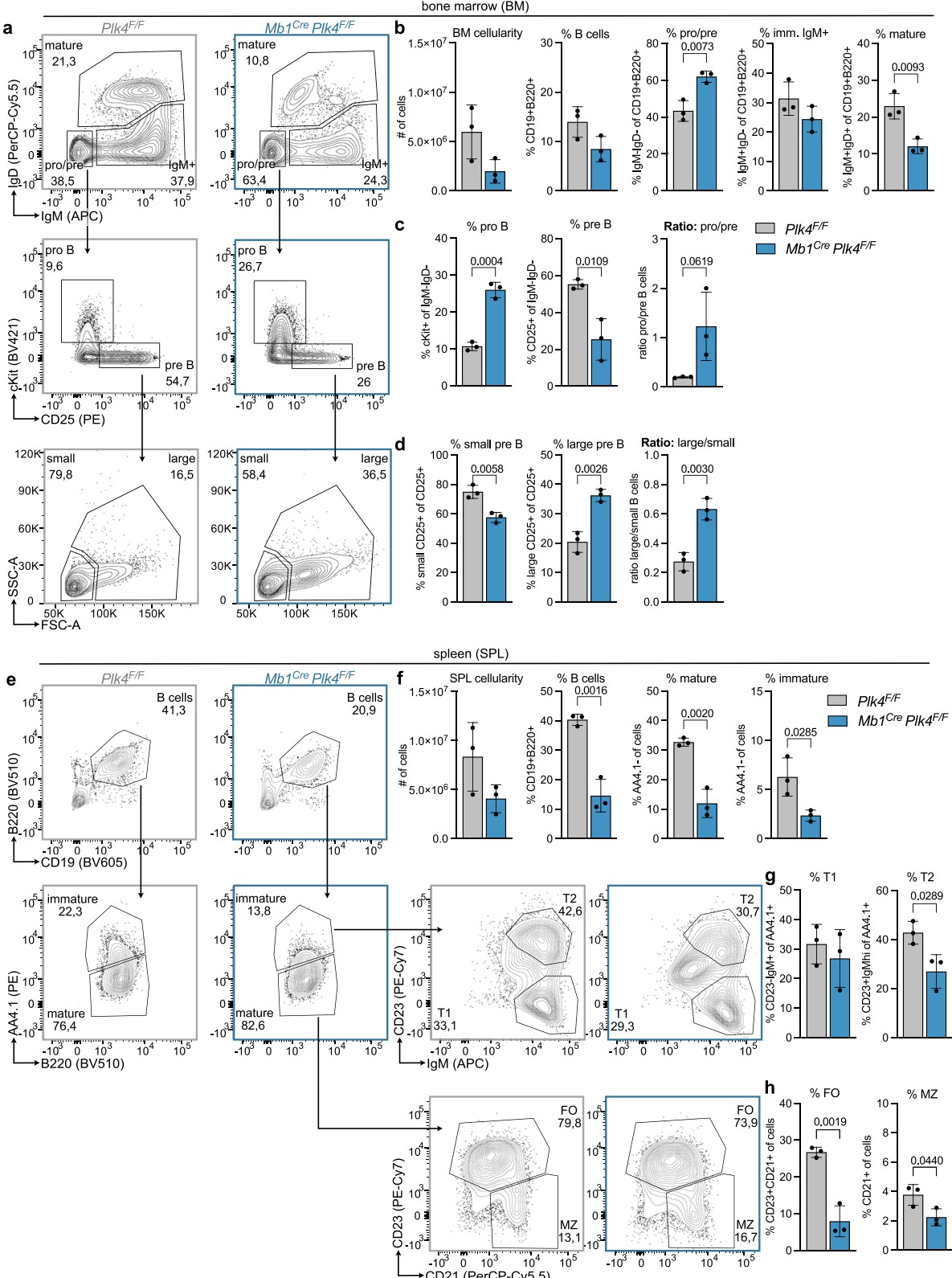

increases tolerance to excessive centriole counts, but that alternative mechanisms are engaged to clear such cells in vivo during B cell ontogeny. Clearly, extra mature centrosomes can promote BCL2-regulated apoptosis in mouse and human hematopoietic cells[61,62]. This phenomenon has recently been assigned to caspase-2´s capacity to selectively proteolyse the BCL2 family protein BID and MDM2 in response to centrosome amplification[62]. At the moment, we can only

speculate that other cell death types may kick in to clear such cells in the bone marrow during differentiation or that extra centrioles themselves are selectively cleared by mechanisms the remain to be defined.

In contrast, the loss of centrioles induced by chemical inhibition of PLK4, leading to the activation of the mitotic surveillance pathway, also triggers BCL2-regulated apoptosis ex vivo that is strictly p53

**Fig. 5 | Loss of PLK4 arrests B cell development. a** Representative FACS plots of control (*Plk4^{F/f}*) and *Mb1^{Cre} Plk4^{F/f}* mice illustrating the gating of pro/pre B (IgD-IgM −), immature (IgD-IgM +) and mature B cells (IgD+IgM +) in the bone marrow after pregating on CD19 + B220 + cells; from pro/pre gate further gating on pro B (cKit + CD25-) and pre B (cKit-CD25 +) cells; and from pre B further gating on small and large pre B cells via FSC-A and SSC-A. **b** Bar graphs show bone marrow cellularity and a fraction of B cells, pro/pre B, immature IgM +, and mature B cells. **c** Bar graphs show a fraction of pro and pre and the ratio of pro to pre B cells within the pro/pre B cell population. **d** Bar graphs show the fraction of small and large pre B and the ratio of large to small pre B cells within the pre B cell population. **e** Representative FACS plots of control (*Plk4^{F/f}*) and *Mb1^{Cre} Plk4^{F/f}* mice illustrating

the gating of splenic B cells (CD19 + B220 +); immature (AA4.1 +) and mature B cells (AA4.1-); from immature B cell gate further gating on transitional 1 (T1, IgM+CD23-) and transitional 2 (T2, IgM+CD23 +) B cells and from mature B cell gate further gating on follicular (FO, CD23 +) and marginal zone (MZ, CD21 + CD23-) B cells; **f** Bar graphs show spleen cellularity and fraction of B cells, mature and immature B cells in the spleen. **g** Bar graphs show the fraction of transitional T1 and T2 cells within the population of immature B cells. **h** Bar graphs show the fraction of follicular (FO) and marginal zone (MZ) B cells within the population of all cells. Error bars in all panels represent mean ± SD. *n* = 3 biological replicates for both genotypes (individual mice). Compared to the Student's two-tailed, unpaired *t* test; Source data are provided as a Source Data file.

dependent (Fig. 4). This is surprising, given the proposed developmental repression of p53 in progenitor B cells. Clearly, mitotic delays after centrinone treatment lead to p53 accumulation and transcriptional activation of its pro-apoptotic target genes, including *Bax*[63], *Puma*, and *Noxa*[64,65], promoting BCL2-regulated apoptosis[66]. Consistent with the detrimental effects of PLK4 inhibition on progenitor B cell survival, we noted developmental arrest of B cells at the pro B cell stage upon *Mb1^{Cre}*-driven deletion of *Plk4*, correlating with increased cell death ex vivo and leading to a severe drop in the percentage and number of all subsequent B cell subsets in bone marrow and spleen of *Mb1^{Cre}Plk4^{f/f}* mice (Fig. 5 and Supplementary Fig. S5). This cell loss could be largely corrected by co-deletion of *Usp28*, leading to restoration of B cell numbers in bone marrow and spleen (Fig. 6). Our findings are in line with studies demonstrating the rescue of neuronal progenitor cells lost due to activation of the mitotic surveillance pathway by co-deletion of *Usp28* or *p53*[15]. Loss of *Usp28* was also shown to rescue development in early mouse embryos lacking the centriolar assembly factor SAS4[14,16]. Conditional deletion of *Sas4* revealed a similar cell death response during lung development[67]. Together, these findings place progenitor B cells on the list of cell types that critically depend on the surveillance of exact centriole counts and that respond with apoptosis upon their loss. This feature appears far from universal, as centriole deficiency does not affect developmental, neonatal, or compensatory hepatocyte proliferation in mice lacking *Plk4* in the liver[32], nor that of gastrointestinal stem cells lacking *Sas4*[67]. Mechanisms underlying these cell type or tissue-dependent differences remain to be uncovered, but the relative expression levels of *p53* or gene locus accessibility may be part of the answer.

Loss of *Usp28* expression not only restored progenitor B cell survival and maturation, but also mature B cell function, as indicated by immunoglobulin production in steady state, but also upon OVA-challenge (Fig. 6c). While the loss of *Plk4* did not significantly affect IgM immunoglobulin levels, representing largely natural antibodies that are produced in the absence of infection, IgG1 levels clearly dropped in *Plk4* mutant mice, but were back to normal when *Usp28* was co-deleted and B cell maturation restored (Fig. 6c). The latter suggests the differential impact of *Plk4*-deficiency on fetal liver-derived B1 B cells that produce most IgM found in steady state and conventional B2 B cells that mature in the bone marrow. Our OVA-immunization experiment indicates that acentrosomal B cells can undergo class switch recombination (CSR), producing comparable levels of OVA-specific IgG1, and hence must be able to expand and survive in the absence of centrosomes when p53 signaling is blunted. Similar findings have been made in NPCs carrying mutations affecting centriole biogenesis and mitotic timing[15]. While progenitor B cells appear to share this capacity, survive, and mature in the absence of centrioles when the mitotic surveillance pathway is blocked, this does not seem to be a prerequisite for follicular B cell expansion. Our findings in iGC B cell cultures in the presence of centrinone support the notion that mature B cells do not rely on the presence of centrioles for their activation, proliferation, or CSR (Supplementary Fig. S6). This feature likely relies on the active suppression of a p53 response by

BCL6 in GC B cells[53], supported by our observations of impaired p53 target gene induction in response to centrinone treatment (Fig. 6). Proliferation in the absence of centrioles may further be facilitated by the low-level expression of *Trim37* in mature vs. progenitor B cells, as this E3 ligase was reported to antagonize acentriolar cell division increasing cancer cell vulnerability to centrinone treatment[50,51]. This finding highlights differences in the efficacy of p53 repression in developing progenitor vs. activated B cells exposed to centrinone (Figs. 4, 6).

Given the proposed role of the centrosome in lysosome positioning for antigen extraction at the immunological synapse in B cells[68], it was interesting to see that *Mb1^{Cre}Plk4^{F/F}Usp28^{F/F}* mice appear to mount a normal IgG1 response to immunization (Fig. 6c). If these immunoglobulins are equal in affinity to OVA-specific antibodies produced in control animals was not assessed. Regardless, our IF analysis of CD19^+ positive B cells from the blood documents the absence of centrosomes in the residual B cells found in the absence of *Plk4*. Notably, the few remaining centrioles also displayed structural defects (Fig. 6), and hence were unlikely fully functional. This argues against counterselection and expansion of residual B cells that may have escaped homozygous *Plk4* deletion. As such, we propose that mature B cells experiencing centriole depletion can still proliferate and expand upon antigenic challenge in vivo.

Together, our results suggest that B cells are intolerant to centriole loss but permissive to centriole accumulation. While the cause for this permissiveness for centriole accumulation nor their potential physiological function in progenitor B cells are not yet known, it is tempting to speculate that the malignant transformation of these cells, most frequently seen in early childhood, may be facilitated by the survival of cells carrying extra centrioles. Open questions to address in future studies are how extra centrioles are cleared during B cell ontogeny in vivo and if the absence of centrosomes may affect antibody affinity maturation, given their documented role in antigen extraction and presentation on B cells.

## Methods

### Ethics statement and animal models

Breeding colonies were approved by the Austrian Federal Ministry of Education, Science and Research (BMWF: 66.011/0008-V/3b/2019), or approved by the Johns Hopkins University Institute Animal Care and Use Committee (MO21M300). Generation and genotyping of *R26-rtTA*, *TET-Plk4*[48], *EGFP-CENT1*[69], *Vav-BCL2^{tg45}*, *Vav-Mcl1^{tg46}*, *p53^{-/-70}*, *Pidd1^{-/-71}*, *Rag1^{-/-72}* and *p21^{-/-73}* mice were described before. These mice were backcrossed and maintained on a C57BL/6 N background for at least 12 generations and housed at the animal facility of the Medical University of Innsbruck under SPF conditions. *Usp28^{F/-}* mice were obtained from the laboratory M. Eilers[74]. *Plk4^{F/F}* mice were generated as described previously[75] and are available at The Jackson Laboratory Repository (Stock #037549). *Mb1^{Cre}* mice were obtained from The Jackson Laboratory Repository (Stock #020505). These mice were kept on a mixed background and were housed and cared for in an AAALAC-accredited facility. For immunization, mice were injected

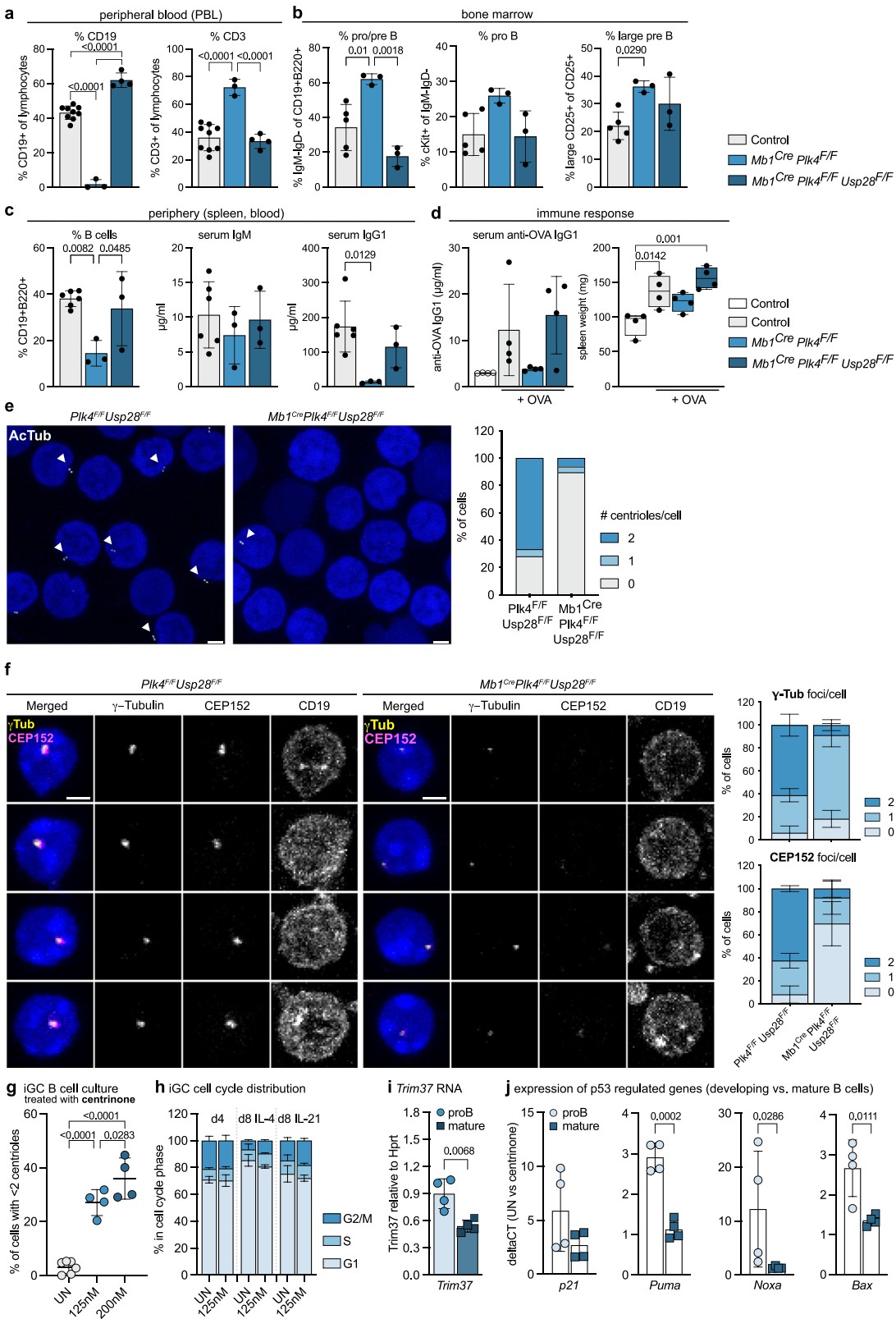

i.p. with 100 µg Endofit Ovalbumin (Invivogen, San Diego, CA, USA, 9006-59-1) mixed in a 1:1 ratio with Imject™ Alum Adjuvant (Thermo Fisher, Waltham, MA, USA, 77161) in a volume of 200 µl per mouse. All animals were kept under standard housing conditions, with a 12 h/ 12 h dark/light cycle with 21 °C ambient temperature and 50% humidity. For isolation of pro B cells, mice were used at the age of 5 weeks (± two weeks), for all other experiments, adult mice with

3–5 months of age were used. For all experiments, male and female littermate mice were used indiscriminately, except for *p53*−/− animals, which were only male.

### Preparation of single-cell suspensions
For the generation of single-cell suspensions murine organs (spleen, thymus) were meshed through 70 µm filters (Corning, Cambridge, MA,

**Fig. 6 | Co-depletion of USP28 restores B cell development in the absence of PLK4. a** Percentage of CD19+ or CD3+ lymphocytes in the peripheral blood (PBL) of *Plk4^{F/F}* or *Mb1^{Cre}* (*n* = 9, pooled controls), *Mb1^{Cre} Plk4^{F/F}* (*n* = 3) and *Mb1^{Cre} Plk4^{F/F} Usp28^{F/F}* mice (*n* = 4), determined by flow cytometry. **b** Fraction of pro/pre B, pro B, and large pre B cells in bone marrow. Control (*Plk4^{F/F}* or *Plk4 Usp28^{F/F}*) = 5, *Mb1^{Cre} Plk4^{F/F}* *n* = 3, *Mb1^{Cre} Plk4^{F/F} Usp28^{F/F}* *n* = 3. **c** Fraction of splenic B cells of the genotypes listed in (**b**) was determined by flow cytometry. Serum IgM and IgG1 levels were determined by ELISA. **d** Serum anti-OVA-IgG1 and spleen weight of indicated genotypes were analyzed 13 days post-immunization. *n* = 4, as control homozygous and heterozygous *Plk4* and *Usp28^{F/F}* mice were pooled. **e** Expansion microscopy (by factor 4) to determine centriole numbers in splenic B cells with acetylated Tubulin (AcTub) antibody and DAPI staining. Scale bars represent 2.5 µm. *n* = 96 cells per genotype. **f** Representative images (left) and number of CEP152 or γ-Tubulin foci (right) in CD19 + B cells of PBL by IF with CD19, CEP152, γ-Tubulin antibody and DAPI nuclear staining. *n* = 3. **g** Centriole depletion assessed by IF with γ-Tubulin, CP110 antibodies, and Hoechst staining in iGC B cells cultured on feeder cells with IL-4 for 4 days ± centrinone. *n* = 6 untreated (UN), *n* = 4 treatment. **h** DNA content analysis of iGC B cells (IL-4 for 4 days, + 4 days with either IL-4 or IL-21). *n* = 3. **i** *Trim37* levels relative to *Hprt* were assessed by qRT-PCR in pro B and LPS-activated mature splenic B cells treated with 125 nM centrinone plus QVD. Data are shown as mean ± SD. *n* = 4. **j** qRT-PCR analysis of p53 target genes in untreated versus centrinone-treated pro B or LPS-treated splenic B cells. *n* = 4. Error bars in all panels represent mean ± SD (except error bars for spleen weight depict min to max) of biological replicates (individual mice). One-way-ANOVA Tukey's multiple comparisons test (**a**–**d**, **g**) and Student's two-tailed, unpaired *t* test (**i**, **j** except for *Noxa* two-tailed Mann-Whitney test was employed due to non-normal data distribution). Source data and exact *p*-values (< 0.0001) are provided as a Source Data file.

USA, 352350), bone marrow was harvested by flushing both femurs and tibiae using a 23 G needle with staining buffer, consisting of PBS with 2 % FBS (Gibco, Grand Island, NY, USA, 10270-106) and 10 µg/ml Gentamycin (Gibco, 15750-037). Peritoneal lavage was performed post-mortem by injecting 10 ml staining buffer in the peritoneal cavity using a 27 G needle, gently massaging the peritoneum, and extracting the fluid again with a 27 G needle. Erythrocyte depletion was performed by incubating cells for 3 min in 1 ml lysis buffer (155 mM NH4Cl, 10 mM KHCO3, 0.1 mM EDTA; pH7.5) on ice. Cells were washed with staining buffer and filtered through a 50 µm cup falcon (BD Biosciences, San Diego, CA, USA, 340632). Cell numbers were determined using a hemocytometer and trypan blue exclusion.

### Cell sorting and flow cytometry

Single-cell suspensions were preincubated with 1 µg/ml of aCD16/32 Fc-Block (BioLegend, San Diego, CA, USA, 101310) in 300 µl staining buffer for 10 min, washed and stained for 15 min with antibodies in a volume of 300 µl staining buffer. The sorted cell subsets were defined as follows: **Bone Marrow:** pro B cells (B220loCD19 + IgD-IgM-CD25-cKit +), large pre B cells (B220loCD19 + IgD-IgM-CD25+ckit-FSChi), small pre B cells(B220loCD19 + IgD-IgM-CD25+ckit-FSClo), immature IgM + B cells (B220loCD19+IgD-IgM +), CLP (Lin-cKit^{int}Sca1 + CD127^{hi}) and LSK (Lin-cKit + Sca1 +) (Supplementary Figs. S7e, S8a). **Spleen:** Follicular B cells (B220 + CD19 + AA4.1-IgM +CD21 + CD23 +), marginal zone B cells (B220 + CD19 + AA4.1-IgM +CD21 + CD23-), transitional B cells (CD19 + B220 + AA4.1 +), CD4/8 naïve (CD3 + CD4 +/CD8 + CD62L + CD44-), CD4/8 CM (CD3 + CD4 +/CD8 + CD62L-CD44 +), CD4/8 EM (CD3 + CD4 +/CD8 + CD62L + CD44 +) (Fig. S8b,e). **Thymus:** DP (CD4 + CD8 + LIN-), DN (CD4-CD8-LIN-)(Fig. S8d). **Peritoneal Cavity:** B1 B cells (B220loCD19 + CD43 +) (Supplementary Fig. S8c). Cell sorting was carried out on a FACS Aria III (BD Biosciences). Non-singlet events were excluded from analyses based on characteristics of FSC-H/FSC-W and SSC-H/SSC-W (Supplementary Figs. S7e, S8a–e).

For the analysis of cultured cells by flow cytometry, the staining procedure was performed in 96-well plates, and cells were washed with 200 µl staining buffer by centrifugation at 500 g for 2 min. Staining was conducted on ice or at 4 °C. To block non-specific antibody-binding, cells were preincubated with 1 µg/ml of αCD16/32 Fc-Block (BioLegend, 101310) in 30 µl staining buffer (PBS, 2 % FCS, 10 µg/ml Gentamycin) for 10 min, washed, and stained for 20 min with 30 µl antibody cocktails. The following fluorescent-labeled anti-mouse antibodies were used at dilution 1:100 – 1:1000

| Channel | | Provider | Catalog # | Clone | Dilution |
|---|---|---|---|---|---|
| FITC | αIgD | Biolegend | 405704 | 11-26 c.2a | 1:100 |
| | αCD19 | Biolegend | 115506 | 6D5 | 1:100 |
| | αIgG1 | eBioscience | 553443 | A85-1 | 1:100 |
| APC | αCD93 (AA4.1) | Biolegend | 136510 | AA4.1 | 1:100 |
| | αcKit | Biolegend | 105812 | 2B8 | 1:100 |
| | αCD11b | eBioscience | 17-0112-83 | M1/70 | 1:200 |
| | αCD25 | Bioloegend | 101910 | 3C7 | 1:300 |
| | αCD19 | Biolegend | 115512 | 6D5 | 1:100 |
| | αCD62L | BD | 553152 | MEL-14 | 1:100 |
| | Streptavidin | Biolegend | 405207 | | 1:400 |
| | αIgM (μ-chain) | Jackson ImmunoResearch | 115-607-020 | Daylight/ AF647 | 1:1000 |
| PE | αCD25 | Biolegend | 101904 | 3C7 | 1:200 |
| | αB220 | BD | 553090 | RA3-6B2 | 1:1000 |
| | αCD93 (AA4.1) | eBioscience | 12-5892-82 | AA4.1 | 1:200 |
| | αIgM | Biolegend | 406507 | RMM-1 | 1:100 |
| | αCD62L | Miltenyi Biotec | 130-112-836 | REA828 | 1:200 |
| | αSca1 | Biolegend | 108107 | D7 | 1:400 |
| | αphH3 | Biolegend | 650807 | 11D8 | 1:200 |
| | Streptavidin | eBioscience | 12-4317-87 | | 1:400 |
| PECy7 | αCD23 | eBioscience | 25-0232-82 | B3B4 | 1:100 |
| | αCD3 | eBioscience | 25-00331-82 | 145-2C11 | 1:100 |
| | αCD127 | Biolegend | 135013 | A7R34 | 1:100 |
| | αB220 | Biolegend | 103222 | RA3-6B2 | 1:200 |
| | αCD117 (cKit) | Biolegend | 105814 | 2B8 | 1:200 |
| | αIgM | Biolegend | 406514 | RMM-1 | 1:200 |
| PerCP-Cy5.5 | αIgD | Biolegend | 405710 | 11-26 C.2 A | 1:300 |
| | αCD21/CD25 | Biolegend | 123415 | 7E9 | 1:1000 |
| | αpH2AX | eBioscience | 46-9865-42 | CR55T33 | 1:400 |
| | αCD4 | eBioscience | 45-0042-82 | RM4-5 | 1:400 |
| APC-Cy7 | αCD4 | BD | 552051 | GK1.5 | 1:100 |
| A700 | αIgD | Biolegend | 405730 | 11-26 c.2a | 1:100 |
| | αCD21/CD35 | Biolegend | 123431 | 7E9 | 1:100 |
| | αCD4 | Biolegend | 116021 | RM4-4 | 1:100 |
| | αB220 | Biolegend | 103232 | RA3-6B2 | 1:400 |
| | αCD11b | Biolegend | 101222 | M1/70 | 1:200 |
| | αIgG1 | Biolegend | 406632 | RMG1-1 | 1:100 |
| | αCD3e | BD | 557984 | 500A2 | 1:100 |
| Brilliant Violet 421 | αCD117 (cKit) | Biolegend | 105828 | 2B8 | 1:200 |
| | αCD8a | Biolegend | 100738 | 53-6.7 | 1:100 |
| | αCD5 | Biolegend | 100617 | 53-7.3 | 1:100 |
| eFluor 450 | αIgM | eBioscience | 48-5890-82 | eB121-15F9 | 1:100 |

| Brilliant Violet 510 | αB220 | Biolegend | 103247 | RA3-6B2 | 1:200 |
| | αCD44 | Biolegend | 103044 | IM7 | 1:100 |
| | αCD138 | Biolegend | 142521 | 281-2 | 1:200 |
| | αSca1 | Biozym | 108129 | D7 | 1:100 |
| Brilliant Violet 605 | αCD19 | Biolegend | 115540 | 6D5 | 1:200 |
| | Streptavidin | Biolegend | 405229 | | 1:400 |
| Biotinylated | αB220 | Biolegend | 103204 | RA3-6B2 | 1:200 |
| | αIgE | BD | 553419 | R35-118 | 1:200 |
| | αCD19 | Biolegend | 115504 | 6D5 | 1:200 |
| | αCD11b | Biolegend | 101204 | M1/70 | 1:200 |
| | αTCRb | Biolegend | 109204 | H57-597 | 1:200 |
| | αTer119 | Biolegend | 116204 | Ter119 | 1:200 |
| | αNK1.1 | Biolegend | 108704 | PK136 | 1:200 |
| | αGr1 | Biolegend | 108404 | RB6-8C5 | 1:200 |

Biotin-conjugated antibodies were incubated with fluorophore-labeled streptavidin conjugates (1:400) before flow cytometric analysis. A minimum of 50.000 cells were acquired. Annexin V-FITC (1:1000, Biolegend 640945) and TO-PRO™-3 Iodide 642/661 (1:50.000, Thermo Fisher, T3605) were added to the cells directly before flow cytometric analysis diluted in Annexin V Binding Buffer (1:10 in water, eBioscience, USA, 00-0055-56) and then used to assess cell death (gating, see Supplementsry Fig. S7a). Cells were then measured on a flow cytometer (LSRII-Fortessa, BD Biosciences), and data were analyzed quantitatively, excluding doublets, using FlowJo (FlowJo X, LLC). All gating strategies can be found in Supplementary Figs. 7 and 8.

## DNA content analysis

Fixation was performed in 200 μl ice-cold 70 % ethanol for a minimum of 16 h at − 20 °C in 96-well plates. Prior to antibody staining, cells were washed twice, adding 200 μl PBS (500 g., 5 min) to remove ethanol, transferred, and subsequently permeabilized with 200 μl PBS containing 0.25 % Triton X-100 (Sigma-Aldrich, St. Louis, MO, USA, T8787) for 20 min. Cells were incubated with γH2AX (Cell Signaling, Beverley, MA, USA, 2577, 1:400) and phospho-Histone H3 Ser-10 (Cell Signaling, 9701, 1:400) antibody in 30 μl permeabilization buffer for 20–30 min. Cells were washed twice with permeabilization buffer, resuspended in 100 μl PBS containing 250 μg/ml RNase A (Sigma R5500), and incubated for 20 min at 37 °C. Finally, 50 μl of 3 μM TO-PRO™-3 Iodide 642/661 (Thermo Fisher, T3605) or 50 μl 10 μg/ml DAPI (Sigma-Aldrich, D9542) in PBS was added. A minimum of 10.000 cells was acquired at LOW mode. Data were acquired on an LSRII cytometer (BD Biosciences) and analyzed using FLOWJO software (Tree Star, Ashland, OR, USA). Non-singlet events were excluded from analyses using FSC-H/FSC-W and SSC-H/SSC-W characteristics, and in case of DNA staining in addition to DAPI 405-A/405-H or TO-PRO-3 APC-A/APC-H (Supplementary Fig. S7b).

## Pro B cell culture

To expand pro B cells ex vivo, 2–4 × 10⁵ B220^lo CD19^+ IgD^- IgM^- CD25^- cKit^+ cells were FACS-sorted from the bone marrow of approximately 5-week-old mice. One-third of the cells were used for cell cycle analysis and centriole count, the remaining cells were then plated in 96-well plates in B cell Medium (DMEM medium (Sigma-Aldrich, D6429) supplemented with 10 % FBS, 2 mM L-glutamine (Sigma-Aldrich, G7513), 0.055 mmol/l 2-ME (Thermo-Fisher, 31350010), 10 mmol/l HEPES (Sigma-Aldrich, H0887), 1 mmol/l sodium pyruvate (Gibco, 11360-039), 1 x NEAA (Gibco, 11140-035), 100 units/ml penicillin, and 100 μg/ml streptomycin (Sigma-Aldrich, P0781) together with 20 ng/ml IL-7 (Peprotech, Rocky Hill, NJ, USA, 217-17). A separate 96-well was prepared for the analysis of each day/condition. Cells were harvested by pipetting, washed once with PBS, and further processed depending on the analysis. Doxycyclin (Sigma-Aldrich, D9891, 1 μg/ml) was added to the culture medium 48 h after plating,

while DHCB (Sigma-Aldrich, D1641) was added after 72 h in culture with IL-7. Centrinone (LCR-263, MedChemExpress, Monmouth Junction, NJ, USA, HY-18682) and CHK1 inhibitor (CHK1i, PF-477736, Selleckchem, Houston, Texas, USA, S2904) were added directly after plating the cells. Concentrations of both compounds are indicated in the figure legends. For differentiation experiments, BCL2^tg pro B cells were expanded 3 days with IL-7, then washed three times with PBS, and then plated in B cell Medium without IL-7. 500 nM MG132 (Sigma-Aldrich, C2211) and 1 μM Chloroquine (Sigma-Aldrich, C6628) were added directly after plating in IL-7 deprived Medium.

## OP9 pre B cell culture

To expand pre B cells ex vivo, 7 × 10⁶ total bone marrow cells from EGFP-CENT1 mice were plated on 6 × 10⁵ OP9 stromal cells, pretreated with 10 μg/ml mitomycin C (Sigma, M0305) in a 15 cm dish. Cells were plated in 15 ml pre B cell Medium (IMDM medium (Sigma-Aldrich, I3390) supplemented with 10 % FBS, 2 mM L-glutamine (Sigma-Aldrich G7513), 0.055 mmol/l 2-ME (Thermo Fisher 31350010), 1 x NEAA (Gibco, 11140-035) and 100 units/ml penicillin together with 10 ng/ml IL-7 (Peprotech, 217-17). After 4 days, cells are collected and all plated again on mitomycin C-treated OP9 stromal cells. One week after the initial plating pre B cells were harvested, stained with CD19-BV605, CD11b-A700 and TO-PRO3 (1:50.000) and subjected to FACS-sorting according to low (30 %), medium (30 %) or high (30 %) EGFP-CENT1 intensity. Cells were then directly used for flow cytometric analysis. For the analysis of SYK inhibition, pre B cells were collected after 1 week of expansion and plated for 48 h with IL-7 and different concentrations of the SYK inhibitor, R406 (Selleckchem, S2194).

## Induced Germinal Center culture (iGC) B cell culture and stimulation of mature B cells

B cells were isolated from splenic single-cell suspension using Magni-Sort Streptavidin Negative Selection Beads (Thermo Fisher Scientific, MSNB-6002-74) and biotinylated antibodies against Ter119 (BioLegend, 116204, 1:100), CD11b (BioLegend, 101204, 1:100) and TCRβ (BioLegend, 109204, 1:50). The iGC B cell culture was performed as described by Nojima et al. [52]. Briefly, 4 × 10⁵ 40LB feeder cells were treated with 10 μg/ml mitomycin C (Sigma-Aldrich, M0305) in 1 ml DMEM (Sigma-Aldrich, D6429) supplemented with 10 % FBS (Gibco, 10270-106), 2 mm l-glutamine (Sigma-Aldrich, G7513) and 100 μ/ml penicillin/100 μg/ml streptomycin (Sigma-Aldrich, 0781). Cells were washed five times with PBS and then, 7,5 × 10⁵ B cells were added in B cell medium: DMEM supplemented with 10 % (v/v) FBS, 2 mM l-glutamine, 10 mM Hepes (LONZA, Basel, Switzerland, BE17-737E), 1 mM sodium pyruvate (Gibco, 13360-039), 1 × nonessential amino acids (Gibco, 11140-035), 100 μ/ml penicillin/100 μg/ml streptomycin, 50 μM β-mercapto-ethanol (Sigma-Aldrich, M3148), 10 ng/ml rIL-4 (Peprotech, 214-14) and treated with different concentrations of centrinone. On day 4, iGC B cells were harvested by collecting the medium and washing plates with harvest buffer (PBS with 0.5 % BSA and 2 mM EDTA). Cells were either analyzed or 7.5 × 10⁵ iGC B cells were replated per 6-well containing fresh mitomycin C treated 40LB feeder cells as detailed above and cultivated in 4 ml of B-cell medium containing either 10 ng/ml rIL-4 or 10 ng/ml rIL-21 (Peprotech, 210-21). On day 8, B cells were harvested and used for flow cytometric analysis.

For surface flow cytometric analysis. iGC B cells were stained as described in the flow cytometry section with the following antibodies (B220 PE, IgM PECy7, IgD PerCPCy5.5, CD22 APC, IgG1 A700, CD19 BV605, CD138 BV510). Before the acquisition, cells were labeled with Fixable Viability Dye eFluor 780 (Thermo Fisher, 65-0865-14) as per the manufacturer's instructions (Supplementary Fig. S7c). For intracellular flow cytometric analysis, iGC B cells were washed once with PBS and then treated with trypsin for 10 min at 37 °C. Subsequently, cells were washed with staining buffer and fixed by the addition of self-made fixation solution (PBS + 4 %PFA + 0,1 % Saponin) for 20 min at 4 °C.

Cells were washed two times with Perm/Wash (PBS + 1 % BSA + 0,1 % Saponin + 0,025 % NatriumAzid) and then incubated with the following antibodies for 15 min (IgG1 FITC, pH3 PE, IgM PECy7, gH2AX PerCPCy5.5, IgE Bio). After washing with perm/wash buffer, cells were incubated with a second antibody solution (Strep BV605) and, after 15 min incubation further processed as described in the section intracellular staining and DNA content analysis (Supplementary Fig. S7d). For the stimulation of mature B cells, B cells were isolated via negative selection from the spleen as described earlier. $4 \times 10^5$ B cells were plated in a 96 well-plate stimulated with LPS (Sigma-Aldrich, L2880, 1 μg/ml) and treated with 125 nM Centrinone. After 48 h, B cells were collected and used for qRT-PCR.

### Immunofluorescence microscopy

For the imaging of B cells from peripheral blood, blood was collected from the submandibular vein into EDTA-coated tubes (BD Biosciences, 365974), and red blood cells were lysed in RBC Lysis Buffer (eBioscience™, 00-4333-57) for 5 min at room temperature. Tubes were centrifuged at 600 g for 5 min, and pellets were washed in PBS 2 % FBS, before incubation with CD19 Alexa Fluor 647 (BioLegend, 115522, 1:100) for 30 min at room temperature.

In general, cells were washed once with PBS and transferred to poly-L-lysine (Sigma-Aldrich, P8920) coated coverslips, followed by incubation for 30 min at 4 °C or poly-L-lysine (Sigma-Aldrich, P6407) coated coverslips, followed by incubation at 37 °C for 1 h. Cells were then fixed with 100 % ice-cold methanol for 7 min at − 20 °C and washed 3 times with PBS. Cells were stored for up to two weeks in PBS or stained immediately. Cells were incubated with blocking solution (2.5 % FBS, 200 mM glycine, 0.1 % Triton X-100 in PBS or 2 % bovine serum albumin (GE Healthcare or Calbiochem), 0.01 % Tween 20 (National Diagnostics or Roth) in PBS) for 1 h at room temperature before staining with primary antibodies in blocking solution for 1 hour. Cells were washed 3 x with washing buffer (PBS 0.1 % Triton X-100 or PBS with 0.2 % Tween) and then incubated with secondary antibodies. When DAPI was used, it was added 1:1000 in secondary antibody dilution. Cells were again washed 3 times with washing buffer. When Höchst 3342 (Sigma-Aldrich, 23491-52) was used it was added in the last washing step. Coverslips were mounted in ProLong™ Gold Antifade (Thermo Fisher) or homemade mounting solution (23 % polyvinyl alcohol, 63 % glycerol, and 0,02 % NaN₃ in PBS). The following antibodies were used to perform immunofluorescence staining in murine cells: rabbit polyclonal α-CEP152 (homemade[75], 1:1000), goat polyclonal α-γ-Tubulin (homemade[48], 1:1000), rabbit polyclonal α-CEP135 (homemade[75], 1:1000), mouse α-γ-Tubulin (Sigma-Aldrich, T3195, 1:250); rabbit α-CP110 (Protein Tech, 12780-1-AP, 1:500) mouse α-CEP164 (Santa Cruz Biotechnology, sc-515403, 1:500), rabbit α-LC3B (Novus Biologicals, NB100-2220), goat α -mouse IgG AF568 (Thermo Fisher, A11031, 1:1000), goat α -rabbit IgG AF488 (Thermo Fisher, A-11034, 1:1000), donkey α -goat IgG AF555 (Thermo Fisher, A-21432, 1:800), donkey α -goat IgG AF647 (Thermo Fisher, A-21447, 1:800), goat α -mouse IgG AF647 (Thermo Fisher, A-21235, 1:800), goat α -rabbit IgG AF555 (Thermo Fisher, A-21428, 1:800), gaot α -mouse IgG AF555 (Thermo Fisher, A-21127, 1:800).

Thirty to fifty Z-stacks were acquired at room temperature on a Zeiss Axiovert 200 M microscope with an oil immersion objective (Ph3 Plan-Neofluar 100 ×/1.3 oil, 440481, Zeiss) using the acquisition software VisiView 4.1.0.3. Maximum stack projections of z-stacks were performed using the VisiView 4.1.0.3. Images in Fig. 6 and supplementary Fig. 6 were obtained using an SP8 (Leica Microsystems) confocal microscope with a Leica 63 ×1.40 NA oil objective at 0.5 μM z-sections. ImageJ was used to adjust contrast and brightness and perform quantifications. Centriole numbers were evaluated manually by counting CP110+ or EGFP-CENT1 spots per cell if co-localization with γ-Tubulin was given. A minimum of 150 cells was counted for each

condition (except for some conditions where a lot of cell death was expected).

### Ultrastructure expansion microscopy (U-ExM)

B cells were isolated from mouse spleens using EasySep Mouse B Cell Isolation Kit (Stem Cell Technologies, Vancouver, Canada, 19854) and attached to poly-D-lysine coated coverslips at 37 °C for 1 h. U-ExM was carried out as previously described[76]. Coverslips with unfixed cells were incubated in a solution of 0.7 % formaldehyde with 1 % acrylamide in PBS overnight at 37 °C. Gelation was carried out via incubation of coverslips with cells facing down with 35 μl of U-ExM MS composed of 19 % (wt/wt) sodium acrylate, 10 % (wt/wt) acrylamide, 0.1 % (wt/wt) N,N′-methylenebisacrylamide (BIS) in PBS supplemented with 0.5 % APS and 0.5 % TEMED, on Parafilm in a precooled humid chamber. Gelation proceeded for 10 min on ice, and then samples were incubated at 37 °C in the dark for 1 h. A 4 mm biopsy puncher (Integra Lifesciences, Princeton, NJ, USA, 33–34 P/25) was used to create one punch per coverslip. Punches were transferred into 1.5 ml Eppendorf tubes containing 1 ml denaturation buffer (200 mM SDS, 200 mM NaCl, and 50 mM Tris in ultrapure water, pH 9) and incubated at 95 °C for 1 h. After denaturation, gels were placed in beakers filled with ddH2O for the first expansion. Water was exchanged at least two times every 30 min at RT, and then gels were incubated overnight in ddH2O. Next, to remove excess water before incubation with primary antibody solution, gels were placed in PBS two times for 15 min. Primary antibodies were diluted in 2 % PBS/BSA and incubated with gels at 37 °C for 2.5 hours, with gentle shaking. Gels were then washed in PBST three times for 10 min with shaking and subsequently incubated with secondary antibody solution plus DAPI diluted in 2 % PBS/BSA for 2.5 h at 37 °C with gentle shaking. The gels were then washed in PBST three times for 10 min with shaking and finally placed in beakers filled with ddH2O for expansion. Water was exchanged at least two times every 30 min, and then gels were incubated in ddH2O overnight. Gels expanded between 4.0 × and 4.5 ×according to SA purity. The following antibodies were used for expansion microscopy: mouse monoclonal α-acetylated-α-Tubulin (Cell Signaling Technology, 12152, 1:500), rabbit polyclonal α-Centrin (homemade[77], 1:500), Alexa fluor secondaries 1:800 (Thermo Fisher) and DAPI (1:800) was used to image DNA.

### Quantitative real-time-PCR

Total RNA was extracted from snap-frozen cell pellets using the Quick-RNA Micro Prep Kit (Zymo Research, Irvine, CA, USA, R1050) and DNase digestion as per the manufacturer's instructions. RNA concentrations were measured by Nanodrop, and 100 ng of total RNA where subjected to first-strand cDNA synthesis with iScript cDNA Synthesis Kit (Bio-Rad, Hercules, CA, USA, 170-8891). cDNA was amplified using the Luna Universal qPCR Master Mix (New England Biolabs, Frankfurt, Germany, M3003E) as per the manufacturer's instructions. The qRT-PCR was run on a StepOnePlus Real-time PCR system (Applied Biosystems, Foster City, CA, USA). Gene expression of individual mRNAs was normalized to HPRT using the ΔC(t) method. The following primers were used:

*Hprt* F: 5′-GTCATGCCGACCCGCAGTC-3′,
*Hprt* R: 5′-AGTCCATGAGGAATAAAC-3′,
*Plk4* F: 5′-GGAGAGGATCGAGGACTTTAAGG-3′,
*Plk4* R: 5′- CCAGTGTGTATGGACTCAGCTC -3′,
*Plk2* F: 5′-GCTGAAGGTGGGGAGACTTTG-3′,
*Plk2* R: 5′- AGGACTTCGGGGGAGAGATA-3′,
*Plk1* F: 5′- CCCTATTACCTGCCTCACCA-3′,
*Plk1* R: 5′- ACCACCGGTTCCTCTTTCTC-3′;
*Bax* F: 5′-TGAAGACAGGGGCCTTTTTG-3′,
*Bax* R: 5′-AATTCGCCGGAGACACTCG-3′,
*Noxa* F: 5′-GCAGAGCTACCACCTGAGTTC-3′,
*Noxa* R: 5′-CTTTTGCGACTTCCCAGGCA-3′,
*Puma* F: 5′-AGCAGCACTTAGAGTCGCC-3′,

*Puma* R: 5´-CCTGGGTAAGGGGAGGAGT-3´,
*p21* F: 5´-AATTGGAGTCAGGCGCAGAT-3´,
*p21* R: 5´CATGAGCGCATCGCAATCAC-3´,
*Trim37* F: 5´-TCCAAGCTCTGTTGTTTCAGC-3´,
*Trim37* F: 5´-TTCCGCCCAACGACAGTTC-3´,

## ELISA

For total immunoglobulin levels in serum, 96-well enzyme-linked immunosorbent assay plates (Sigma-Aldrich, CLS3590) were coated with 50 µg/ml capture antibody (Southern Biotech, Birmingham, AL, USA, 1010-01) at 4 °C over-night. Plates were washed three times with wash buffer (PBS containing 0.05 % Tween20), blocked with 100 µl per well 1 % BSA in PBS at room temperature for 4 h, and washed three more times with wash buffer. 100 µl per well of mouse serum serially diluted 1:4 in blocking buffer (range 1:800 to 1:160 000) were incubated with coated wells overnight at 4 °C. Plates were washed three times with wash buffer and incubated with 100 µl per well of HRP-conjugated α-mouse IgG1 (Southern Biotech, 1070-05, 1:5000 in 1 % BSA in PBS) or HRP-conjugated α-mouse IgM (Southern Biotech, 1020-05, 1:5000 in 1 % BSA in PBS) for 4 h at room temperature. For detection, 100 µl of ABTS substrate solution per well [200 µL ABTS (Stock: 15 mg/ml in a.d.), 10 ml citrate-phosphate buffer (574 mg citric acid monohydrate in 50 ml a.d.) and 10 µl $H_2O_2$] was incubated for 20 min. Absorbance was measured at 405 nm using a microplate reader (Tecan Sunrise, Männedorf, Switzerland). For OVA-specific serum IgG1 titers, 13 days after immunization with OVA/alum, Cayman's Anti-Ovalbumin IgG1 (mouse) ELISA Kit (Cayman Chemical, Ann Arbor, MI, USA, 500830) was used according to the manufacturer's protocol.

## Statistics & reproducibility

Results are always shown as mean and standard deviation (SD) unless stated otherwise. No statistical methods were used to predetermine sample sizes. Graphs were plotted, and statistical analysis was performed with GRAPHPAD PRISM 10.1.1 software (GraphPad Software, San Diego, CA, USA) using Student's two-tailed, unpaired *t* test when comparing two groups or two-tailed Mann-Whitney test when comparing two groups that are not normally distributed. One-way ANOVA and Tukey post hoc test were used when comparing multiple groups, Two-way ANOVA and Tukey post hoc test when comparing multiple groups over different time points/conditions, and Bonferroni's multiple comparisons test when comparing 2 groups over different time points/conditions. Normality testing was performed before analyzing data with a paired test. All relevant comparisons were made, and non-significant results were not indicated in the figures. The number of biological repetitions (n) is stated in each figure legend, and every experiment was performed at least twice. Differences between groups were considered statistically significant when $P < 0.05$. In figures, asterisks stand for: *$P < 0.05$; **$P < 0.01$; ***$P < 0.001$; ****$P < 0.0001$. The experiments were not randomized, and the investigators were not blinded to allocation during experiments and outcome assessment.

## Reporting summary

Further information on research design is available in the Nature Portfolio Reporting Summary linked to this article.

## Data availability

The datasets generated during and/or analyzed during the current study are available on ZENODO[78]. For all representative FACS plots, fcs-files for a minimum of 3 individual replicates can be found on ZENODO. All gating strategies can be found in Supplementary Figs. S7 and S8. Source data are provided with this paper.

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

## Acknowledgements

We thank C. Soratroi, I. Gaggl and J. Heppke for expert technical assistance, M. Sauerwein for animal care, as well as S. Geley for help with microscopy and critical discussion. EK acknowledges support from the Deutsche Forschungsgesellschaft (DFG), EXC 2151 – 390873048. AJH acknowledges support from the National Institutes of Health, R01GM114119, R01GM133897, and R01CA266199. AV acknowledges support from the Austrian Science Fund (FWF) [https://doi.org/10.55776/DOC82] and the European Research Council, ERC AdG POLICE (787171). For open access purposes, the author has applied a CC BY public copyright license to any author-accepted manuscript version arising from this submission.

## Author contributions

M.S. conducted experiments, analyzed data, prepared figures, and wrote manuscripts. V.Z.B., G.L.M., and M.S.L. conducted experiments and managed mouse colonies. M.H. and M.S.L. generated mouse models used in this study. E.K. contributed to project conceptualization and supervision. A.J.H., V.L., and A.V. designed the research, analyzed data, wrote the manuscripts, and conceived the study.

## Competing interests

The authors declare no competing interest
