## [Peer Review file · Nature Communications]

Centrioles are frequently amplified in early B cell development but dispensable for humoral immunity

Corresponding Author: Dr Andreas Villunger

Version 0:

Reviewer comments:

Reviewer #1

(Remarks to the Author)

In this manuscript, Villunger and colleagues characterize centrosome numbers during B-cell development in mice. They found an abnormally high centriole number specifically in pro-B and large pre-B cells; however, centriole number was back to normal in subsequent differentiated states. The authors demonstrated that loss of cells with amplified centrosomes during differentiation is not due to piddosome, p21, p53 or even due to apoptosis since overexpressing bcl2 does not lead to the maintenance of cells with extra centrosomes past pro-B/large pre-B cell stage. Conversely, complete loss of centrioles, leads to apoptosis of progenitor B-cells, a process that can be restored by the surveillance mechanism previously described that monitors loss of centrosomes. Interestingly, B-cell function in cells without centrosomes is also restored, suggesting that centrosomes are dispensable for B-cell function.

In general, the data presented here is interesting and robust, however the findings are somehow limited by the lack of mechanistic understanding. I appreciate much of the work is done in vivo, and while this is important as we lack fundamental understanding and characterization of centrosome number/structure during differentiation for example, I do not think the works advances significantly our understanding at this stage. I highlight some points below.

1. In Figure 2, the authors test the possibility of Pidd1, p53 and p21 affecting the centriole number in progenitor B-cells by using Pidd1^{-/-}, p53^{-/-} and p21^{-/-} mouse models, with the conclusion that these are not involved. It would be interesting to also investigate whether the Piddosome pathway is activated (MDM2 cleavage?) in cells with extra centrioles, especially in the Plk4 overexpression cells. This could be informative.
2. The progenitor B cells are cultured in vitro the presence of IL-7 to prevent differentiation, and in this scenario the authors show a decrease in centriole numbers with time, and suggest this is a consequence of BCL2-mediated apoptosis, as BCL-2 overexpression reverses this (Figure 2). However, centriole numbers are much lower in BCL-2-overexpressing progenitor B-cells compared to wild type at day 0 of culture (Figure 2D). Thus, this contradicts the authors hypothesis that inhibiting cell death prevents elimination of cells with extra centrioles. It seems that inhibiting cell death leads to further centrosome amplification. In addition, in vivo, overexpression of BCL-2 has no effect in the number of centrioles in the later stages of differentiation (Figure 3F), suggesting it does not play a role in the loss of centrioles during differentiation. Therefore, the role of BCL-2 mediated apoptosis in clearing progenitor B-cells with extra centrosomes and, hence, preventing these extra centrioles to be present in later stages of B-cell differentiation, remains unclear.
3. In relation to the point above, the authors do not assess other possibilities for the loss of cells with extra centrioles during differentiation. Could it be that progenitor B cells with extra centrioles are permissive to these abnormalities, but not capable of differentiating? Do these cells have more DNA damage in vivo or other abnormalities that impair their differentiation? The authors also do not investigate what drives this increase in centriole numbers: could this be a result of an arrest or prolonged G2, as previously described? Or do these cells accumulate because they are permissive to extra centrioles but do not continue the differentiation process? Likewise, it is difficult to predict the impact of these cells in vivo. Are they a 'glitch' in the system or play a role in the differentiation/B-cell function?
4. Depletion of centrioles in progenitor B-cells, using centrinone, arrests B-cell development, a process that can be reverted by USP28 depletion, suggesting that if allowed to survive, centrosome loss has little impact on B-cell differentiation and function. This is interesting but I find it difficult to integrate with the data regarding centrosome amplification.

Reviewer #2

(Remarks to the Author)

In this manuscript, Schapfl et al. characterize the centriole count across B-cell development and propose a mechanism of control for the clearance of cells with aberrant centriole numbers. The authors show the presence of extra centrioles in early B-cell progenitors that are cleared as maturation progresses, by a mechanism independent of PIDD or p53. The study provides new insights into B-cell maturation and the molecular checkpoints controlling its progression. There are some aspects that, if clarified, would solidify the manuscript.

1. The authors show how BCL2 ectopic expression decreases the clearance of cells with extra centrioles (Figure 2D), most likely, because of apoptosis blockage. Based on this data, would you expect to have higher frequency of cells with extra centrioles in d0 from the BCL2^{tg} mice? Please clarify.
2. The authors assess cell death by considering the sub-G1 population. Although the presence of this population can be partially explained by increased apoptosis, this mechanism is a very complex process and should be assessed deeply to be able to use sub-G1/cell death/apoptosis as equivalents. Further, in the abstract, the authors state that when there are “too many and too few centrosomes, mitochondrial apoptosis is engaged to kill”.
3. When using the Plk4 deficient mouse model, the frequency of T1 cells is not affected. Could the authors comment on why this could be happening?
4. When discussing the role of p53 after centrinone treatment, the authors suggest that the results could be explained by an accumulation of p53, which would induce the transcription of its targets. It would be valuable, to support this idea with expression data from these experimental conditions.

Minor comments

1. Figure 3 D and E are not referenced in the main text. Also, references to panels for Fig. S4 are confusing in the text.
2. Centrinone treatment causes a considerable degree of cell death (Fig. S4B) in WT and p21^{-/-}. Would this be a concern for the interpretation of the data?

Version 1:

Reviewer comments:

Reviewer #1

(Remarks to the Author)

I thank the authors for addressing the comments raised during the revision. However, the new data further supports the idea that the presence of these amplified centrosomes could be a glitch due to DNA damage and that cell death does not depend on the PIDDosome activation. I understand the cellular differences between tolerance to centrosome loss however taken together these 2 parts in my opinion seem disconnected. I am afraid I cannot see how this work significantly advances the field.

Regarding the specific comments raised during the revision:

1. The authors' new data suggest that indeed PIDDosome is not activated in these cells with amplified centrosomes, likely due to lack of mature centrioles. This for me suggests that this amplification is not what is driving cell death but rather something else that does not allow these cells to be maintained/progress.
2. The mitochondrial apoptosis the authors propose to limit the survival of cells with extra centrosomes is unlikely to be specific and thus something else could be driving this effect, in particular due to the point 1 above. I appreciate the effort to look into this by the authors but there is no direct evidence supporting that apoptosis is activated by extra centrosomes in this case.
3. While I acknowledge the extra work put into addressing this point, the data supports that centrosome amplification in these cells may in fact be a glitch and a consequence of prolonged G2 arrest due to DNA damage (this is known). Therefore, it is impossible to know what is triggering apoptosis in these cells. I would suggest DNA damage could indeed be the trigger, since no PIDDosome activation is found in these cells and pidd1^{-/-} KO does not rescue loss of cells with extra centrosomes.
4. That is a fair explanation by the authors but my point remains that these 2 parts are not well connected in a cohesive story.

Reviewer #2

(Remarks to the Author)

The authors have addressed all my comments satisfactorily.

Version 2:

Reviewer comments:

Reviewer #2

(Remarks to the Author)

The new data maintain the rigor and quality standards of previous versions of the manuscript.

Point-to-point reply NCOMMS-23-50497-T

Reviewer #1 (Remarks to the Author):

In this manuscript, Villunger and colleagues characterize centrosome numbers during B-cell development in mice. They found an abnormally high centriole number specifically in pro-B and large pre-B cells; however, centriole number was back to normal in subsequent differentiated states. The authors demonstrated that loss of cells with amplified centrosomes during differentiation is not due to piddosome, p21, p53 or even due to apoptosis since overexpressing bcl2 does not lead to the maintenance of cells with extra centrosomes past pro-B/large pre-B cell stage. Conversely, complete loss of centrioles, leads to apoptosis of progenitor B-cells, a process that can be restored by the surveillance mechanism previously described that monitors loss of centrosomes. Interestingly, B-cell function in cells without centrosomes is also restored, suggesting that centrosomes are dispensable for B-cell function.

In general, the data presented here is interesting and robust, however the findings are somehow limited by the lack of mechanistic understanding. I appreciate much of the work is done in vivo, and while this is important as we lack fundamental understanding and characterization of centrosome number/structure during differentiation for example, I do not think the work advances significantly our understanding at this stage. I highlight some points below.

Response: we would like to thank this reviewer for the time taken and describing our work as interesting and robust. We also hope that additional experiments and clarifications will also help to increase our current understanding and provide some mechanistic insight.

1. In Figure 2, the authors test the possibility of Pidd1, p53 and p21 affecting the centriole number in progenitor B-cells by using Pidd1^{-/-}, p53^{-/-} and p21^{-/-} mouse models, with the conclusion that these are not involved. It would be interesting to also investigate whether the Piddosome pathway is activated (MDM2 cleavage?) in cells with extra centrioles, especially in the Plk4 overexpression cells. This could be informative.

Responses: We were very keen to tie the accumulation of extra centrioles to PIDDosome activation, but our genetic experiments did not support involvement. Indeed, it remains possible that the PIDDosome may be engaged and acts in a redundant manner with another yet to be identified signaling pathway to clear cells with extra centrioles. As this referee points out correctly, monitoring MDM2 processing would be ideal to evaluate this, but unfortunately, the antibody at hand only recognizes the processed fragments of human MDM2 (PMID:28130345). However, as we hardly ever found cells with extra mature centrosomes (monitored by CEP164 staining in Figure 1d), the activating cue for PIDDosome formation is actually missing in these cells. This observation also suggests that these extra centrioles never mature. We discuss this issue now in more detail on page 7, end of the first paragraph.

2. The progenitor B cells are cultured in vitro the presence of IL-7 to prevent differentiation, and in this scenario the authors show a decrease in centriole numbers with time, and suggest this is a consequence of BCL2-mediated apoptosis, as BCL-2 overexpression reverses this

(Figure 2). However, centriole numbers are much lower in BCL-2-overexpressing progenitor B-cells compared to wild type at day 0 of culture (Figure 2D). Thus, this contradicts the authors hypothesis that inhibiting cell death prevents elimination of cells with extra centrioles.

It seems that inhibiting cell death leads to further centrosome amplification. In addition, in vivo, overexpression of BCL-2 has no effect in the number of centrioles in the later stages of differentiation (Fig. 3F), suggesting it does not play a role in the loss of centrioles during differentiation. Therefore, the role of BCL-2 mediated apoptosis in clearing progenitor B-cells with extra centrosomes and, hence, preventing these extra centrioles to be present in later stages of B-cell differentiation, remains unclear.

Response: This concern is important, well-justified and led us to rethink. While we still believe that apoptosis regulates the number of progenitor B cells with > 4 centrioles, undoubtedly in vitro, also in vivo, apoptosis may not be involved in their clearance in situ, as suggested by this referee, and indicated by our own data.

In favor of a role of apoptotic cell death limiting the number of progenitor B cells with > 4 centrioles speaks the observation that the lower percentage of such cells in BCL2 transgenic mice in situ still translates into cell numbers that are comparable to that seen in wt animals (new Fig. 3g,h), indicating improved survival upon BCL2 overexpression. Moreover, analysis of pro B cells from vav-Mcl1 transgenic mice phenocopies findings made using BCL2 transgenic cells, i.e., improved survival ex vivo facilitating centrosome amplification (Fig. 2). Moreover, vav-Mcl1 transgenic mice show a percentage of cells with > 4 centrioles in situ that is comparable to wild type mice, but show a clear increase in their absolute number in the bone marrow (new Fig. 3f, h). Together, we believe that mitochondrial apoptosis limits the survival of these cells also in vivo, but it is not critical for their removal during B cell maturation. This is also discussed in detail on page 8 in the results and on the bottom of page 11 in the discussion. Moreover, we have changed the abstract accordingly, removing the quote that "too few and too many centrosomes engage mitochondrial apoptosis"...

3. In relation to the point above, the authors do not assess other possibilities for the loss of cells with extra centrioles during differentiation. Could it be that progenitor B cells with extra centrioles are permissive to these abnormalities, but not capable of differentiating? Do these cells have more DNA damage in vivo or other abnormalities that impair their differentiation? The authors also do not investigate what drives this increase in centriole numbers: could this be a result of an arrest or prolonged G2, as previously described? Or do these cells accumulate because they are permissive to extra centrioles but do not continue the differentiation process? Likewise, it is difficult to predict the impact of these cells in vivo. Are they a 'glitch' in the system or play a role in the differentiation/B-cell function?

Response: Excitingly, this referee is reiterating our own thoughts and questions. All these comments are valid and meaningful and we have asked us these question repeatedly ourselves. While we did not find a satisfactory explanation as to how these cells (or centrioles) are cleared, or whether their differentiation may be prevented, we have made good progress addressing the questions related to DNA damage and how these extra centrioles may arise.

- 1) Indeed, introducing staining for gH2AX as a marker of DNA damage, we noted that cells with > 4 centrioles are also showing higher numbers of foci staining positive for

gH2AX, indicating a correlation between the two phenomena that may be causal (new Fig. 1e). We can also exclude a role for “physiological DNA damage induced by IgH rearrangement, as *Rag1*-mutant cells, unable to rearrange the BCR, show a similar increase in cells with > 4 centrioles (new Fig. 1g).

- 2) Exploring cell cycle behaviour further, we made use of the EGFP-CENT1 reporter system, documenting that (i) the EGFP high fraction contains the biggest portion of pro B cells with > 4 centrioles and (ii) the majority of G2 cells (new Fig. S1d-f). This provides further evidence that extra centrioles are created in G2 cells.
- 3) Replication stress and DNA damage have been both noted to allow CA, due to premature centriole disengagement in G2 (PMID:37773176). In support that this may actually be the case in progenitor B cells, we treated cultures ex vivo with CHK1 inhibitor, that overrides the G2/M checkpoint and shortens time in G2. Under these conditions, the percentage of cells with > 4 centrioles was found to drop (new Fig. 1h). Similarly, reducing proliferation rates of these cells by adding a SYK kinase inhibitor caused a similar effect (new Fig.1 j,k).

Taken all together, we believe that the cause of extra centrioles in progenitor B cells is caused by proliferation stress-related DNA damage, extended time in G2 and premature centriole disengagement. These results are integrated in Figure 1 and S1, presented on page 6 and discussed on page 11.

4. Depletion of centrioles in progenitor B-cells, using centrinone, arrests B-cell development, a process that can be reverted by USP28 depletion, suggesting that if allowed to survive, centrosome loss has little impact on B-cell differentiation and function. This is interesting but I find it difficult to integrate with the data regarding centrosome amplification.

Response: Indeed, it may not come as a surprise that once progenitor B cells manage to progress into the IgM+ naïve B cell stage, the absence of centrioles may not hamper further differentiation. That expansion of mature B cells in response to mitogens is not affected by centrosome loss in ex vivo cultures and that antigen challenge does trigger antibody production and class switching was indeed surprising for us. As such, a clear difference can be found in proliferating progenitor B cells and mature follicular B cells, regarding their dependence on centrosomes for MTOC formation during proliferation. This topic is discussed also with Referee#2. We can offer the following explanation for these differences.

- 1) The expression levels of TRIM37, reported to limit acentriolar mitoses (PMID: 32908304), is higher in progenitor B cells (Fig. 6i), compared to mature follicular cells, rendering the former more vulnerable to centriole loss, triggering rapid apoptosis (Fig. 4d,e).
- 2) Centrinone treatment triggers a strong p53 response in progenitor B cells, but not follicular B cells, as indicated by the (lack of) induction of its target genes, including pro-apoptotic *Noxa*, *Puma* or *Bax* (Fig. 4f, Fig. 6j). The latter may be due to BCL6 mediated p53 repression in activated B cells undergoing germinal center reactions (PMID:15577913).

These findings are presented on page 8/9, as well as on page 10, respectively, and discuss the findings on page 12/13.

Reviewer #2 (Remarks to the Author):

In this manuscript, Schapfl et al. characterize the centriole count across B-cell development and propose a mechanism of control for the clearance of cells with aberrant centriole numbers. The authors show the presence of extra centrioles in early B-cell progenitors that are cleared as maturation progresses, by a mechanism independent of PIDD or p53. The study provides new insights into B-cell maturation and the molecular checkpoints controlling its progression. There are some aspects that, if clarified, would solidify the manuscript.

Response: we would like to thank this reviewer for the time taken to provide critical feedback on our work and attesting us to provide *new insights*, supporting the novelty of our findings.

1. The authors show how BCL2 ectopic expression decreases the clearance of cells with extra centrioles (Figure 2D), most likely, because of apoptosis blockage. Based on this data, would you expect to have higher frequency of cells with extra centrioles in d0 from the BCL2^{tg} mice? Please clarify.

Response: This point is well taken and was also picked up by referee #1. Indeed, one would anticipate that if mitochondrial apoptosis plays a key role in clearing such cells in vivo the baseline should be higher in *vav*-BCL2 mice and that such cells persist during differentiation. We believe that the “baseline” is not set solely by a cell death threshold but also by the proliferative index of these cells. The higher the proliferation rate, the higher the percentage of cells with >4 centrioles. This is also supported, in part, by our new findings using *vav*-MCL1 transgenic mice and re-analysis of original data, converting percentages of these cells into absolute numbers. This shows that the number of pro B cells with CA in BCL2 transgenic mice is comparable to that in wt animals (despite their lower percentage), while the lower base-line in BCL2 transgenic mice correlates with their reduced cell cycle activity, noted before by us and others (PMID:9003774), and indicated by the lower fraction of cells in G2/M (new Fig. 3 g, h). Moreover, *vav*-Mcl1 transgenic mice that do not show this cell cycle phenotype actually show an increased number of these cells in the bone marrow (new Fig. 3h) and a similar cell death protection ex vivo (Fig. 3d). This suggests that apoptosis regulates the number of these cells in the bone marrow and their life-span in vitro, but does not contribute to their clearance during B cell ontogeny (as none of the cell death defective genotypes tested shows an accumulation of these cells at later differentiation stages). As such, we have tuned-down our conclusion that cells with extra centrioles are cleared by apoptosis in vivo and amended the abstract, but only state that cell death inhibition increases tolerance towards centriole amplification in progenitor B cells in situ and ex vivo.

2. The authors assess cell death by considering the sub-G1 population. Although the presence of this population can be partially explained by increased apoptosis, this mechanism is a very complex process and should be assessed deeply to be able to use sub-G1/cell death/apoptosis

as equivalents. Further, in the abstract, the authors state that when there are “too many and too few centrosomes, mitochondrial apoptosis is engaged to kill”.

Reply: As stated above, we have now amended the conclusion in the abstract. Moreover, we have employed additional measures to confirm that the cell death we study is apoptosis, as it can be blocked by caspase-inhibition (Fig. 2e, f), and that the sub-G1 fraction actually correlates well with the percentages obtained TOPRO3/AnnexinV staining (Fig. 2e, 4d, S6c). In addition, we have analyzed cells from MCL1 overexpressing mice, that are also protected from undergoing mitochondrial apoptosis (PMID:20631380).

3. When using the Plk4 deficient mouse model, the frequency of T1 cells is not affected. Could the authors comment on why this could be happening?

Response: we believe this is a misconception, while the percentage of Transitional Type1 B cells appears comparable in spleen between genotypes, their absolute number is also strongly reduced. To indicated this, we have moved splenocyte numbers from the supplement to Fig. 5 and display absolute numbers of T1 and T2 B cells in Fig. S5c.

4. When discussing the role of p53 after centrinone treatment, the authors suggest that the results could be explained by an accumulation of p53, which would induce the transcription of its targets. It would be valuable, to support this idea with expression data from these experimental conditions.

We have this addressed this and were able to highlight that the high sensitivity of progenitor B cells to centrinone treatment correlates well with a strong induction of pro-apoptotic p53 target genes, such as *Noxa*, *Puma* and *Bax*, best visible in the presence of a pan-caspase inhibitor, due to reduced cell death kinetics (Fig. 4e,f). Intriguingly, such a response was not noted in drug-resistant mature follicular B cells (Fig. 6j and S6c). In addition, progenitor B cells express higher levels of TRIM37 (Fig. 6i), which likely also impairs their capacity to undergo acentriolar mitoses, similar to what is seen in cancer cells (PMID:32908304). Results are presented on page 8, 9 and page 10; discussed on page 13.

Minor comments

1. Figure 3 D and E are not referenced in the main text. Also, references to panels for Fig. S4 are confusing in the text.

Thanks for pointing this out – corrected.

2. Centrinone treatment causes a considerable degree of cell death (Fig. S4B) in WT and p21^{-/-}. Would this be a concern for the interpretation of the data?

Reply: Indeed, wt and p21-deficient pro B cells are highly susceptible to centrinone, for reasons highlighted just above. We are unclear though why this should compromise data interpretation, as p53 mutant or BCL2 overexpressing cells are fully protected (Fig. 4e).

Referee #1

I thank the authors for addressing the comments raised during the revision. However, the new data further supports the idea that the presence of these amplified centrosomes could be a glitch due to DNA damage and that cell death does not depend on the PIDosome activation. I understand the cellular differences between tolerance to centrosome loss, however taken together these 2 parts in my opinion seem disconnected. I am afraid I cannot see how this work significantly advances the field.

We want to thank this referee again for the time taken to re-evaluate our revised manuscript. Unfortunately, we are not sure why this referee feels that the two parts of the story are disconnected. We explore the cause and consequence of deregulated centriole number in the B cell lineage, tuning the system towards centriole amplification or loss. The fact that not all data match hypotheses that can be made based on current literature does not make it disconnected or incohesive, but, in our point of view, rather more interesting.

We respectfully disagree with the lack of a “significant advance”. We show for the first time that lack of centrioles induces the “mitotic stop watch” pathway in developing B cells in vivo and that the absence of centrioles does not perturb adaptive immune responses. These findings are striking in their own right. Not only do we demonstrate for the first time that this pathway is relevant in the hematopoietic compartment, but we also show that centrosomes, seen as key structural components of the immunological synapse, are actually dispensable for humoral immunity. Our data forces us to reconsider the role and relevance of centrioles in immunity in general, which sparked quite some excitement with colleagues we shared our data who work in lymphocyte development and adaptive immunity.

Moreover, we also provide an explanation why mature B cells differ in their response to centriole loss, compared to progenitor B cells, i.e., due to the complete silencing of a p53 response in the former and different TRIM37 expression levels between the two maturation states, adding mechanistic insight. This observation is also relevant considering putative treatment of B cell malignancies with PLK4 inhibitors currently in development. Hence, I think it is not really justified to state that our work is neither significant nor novel. In fact, our work should be of broad interest beyond “the field” and be informative for cell biologists, immunologists and those interested in B cell development and B cell cancer alike. Hence, in our humble opinion, it may actually not only advance but also expand “the field”. We are still unclear which field this referee actually refers to.

Also, referring to the concern this may all just be a “glitch”. Is there really something like a “glitch” in nature? Even if so, it indicates a deviation from the “norm” and is of interest. It may be relevant in some settings, or explain how physiology may deviate into pathology. Regardless, such inconsistencies have to be noted and reported first, in order to be recognized and studied before eventually being understood, or dismissed, e.g. as an experimental artefact. This referee rightly argues that the effect of DNA damage on centriole amplification was shown before in model cell lines and hence is not interesting any more. Please let me highlight that this phenomenon has been recognized by cell biologists only when exogenous triggers of DNA damage (PMID:24532022, and references discussed within), or replication stress (e.g. PMID:3773176), were applied, but I do not recall an “endogenous” setting being

reported. That this phenomenon actually happens in the absence of any such exogenous trigger in truly primary B cells, in my point of view, is novel and could also be seen as a finding in support of data generated studying model cell lines, and appreciated as such.

Specific comments raised during the revision by R1:

1. The authors' new data suggest that indeed PIDDosome is not activated in these cells with amplified centrosomes, likely due to lack of mature centrioles. This for me suggests that this amplification is not what is driving cell death but rather something else that does not allow these cells to be maintained/progress.

Highly justified remark, but we already excluded activation of the PIDDosome in the original submission of our manuscript and already explained how we interpret the data available to us. Meaning, as we do not find additional mature Cep164⁺ centrioles (Fig. 1D; 2G), we concluded in version # 1 that a lack of PIDDosome activation is likely due to the fact that these centrioles do not mature, as they may be structurally defective.

However, we have added now new data that shows that progenitor B cells indeed can activate the PIDDosome when mature centrosomes accumulate, i.e., in the context of cytokinesis failure, induced by DHCB treatment. This causes massive cell death in wt cells, but cells lacking PIDD1 are less susceptible to death, reflected in the increased number of viable and highly polyploid cells. Overexpressing BCL2 is superior to loss of PIDD1 in this setting, likely because PIDD1 mutant cells will eventually also die in a PIDD1-independent manner, but this death can still be blocked by BCL2. The corresponding data is now shown in the new version of Figure 2 (panels f-h).

2. The mitochondrial apoptosis the authors propose to limit the survival of cells with extra centrosomes is unlikely to be specific and thus something else could be driving this effect, in particular due to the point 1 above. I appreciate the effort to look into this by the authors but there is no direct evidence supporting that apoptosis is activated by extra centrosomes in this case.

This concern is appreciated. We have down-tuned our conclusions accordingly in all relevant sections of the manuscript. Yet, we do show that extra centrosomes can activate BCL2-regulated apoptosis when PLK4 is overexpressed (Fig. 3C). Furthermore, inhibition of apoptosis due to BCL2 or MCL1 overexpression allows cells with extra centrosomes to survive in culture, allowing them to accumulate additional centrioles, while these structures are gradually lost over time in wt cells, indicating that those cells are less fit. Otherwise, their percentage should not drop over the course of 7 days, but rather stay constant (Fig. 2B). Together, we believe it is fair to say that apoptosis inhibition facilitates the survival of cells with additional centrioles and that extra centrosomes can kill progenitor B cells in a PIDD1-dependent and BCL2-regulated manner.

3. While I acknowledge the extra work put into addressing this point, the data supports that centrosome amplification in these cells may in fact be a glitch and a consequence of prolonged G2 arrest due to DNA damage (this is known). Therefore, it is impossible to know what is triggering apoptosis in these cells. I would suggest DNA damage could indeed be the trigger,

since no PIDDosome activation is found in these cells and pidd1^{-/-} KO does not rescue loss of cells with extra centrosomes.

We do not insist that these extra centrioles are triggering apoptosis, as we would not be able to disentangle this from effects caused by DNA damage being higher in these cells (Fig. 1E). However, the notion that p53 mutant cells are transiently better off (Fig. 2A) supports the idea that DNA damage may contribute to cell death. This is highlighted more in the discussion. Regardless of the actual trigger of cell death, loss of p53 also fails to allow those cells to progress in development (Fig. 3D).

4. That is a fair explanation by the authors but my point remains that these 2 parts are not well connected in a cohesive story.

As pointed out above, we do not feel the same. But this is a very personal opinion and we may not be able to convince this referee otherwise. However, this is simply not the case. We cannot change this, but only point out why we find this is actually even more interesting, as it opens up new questions that can be explored.

The referee also states *“This for me suggests that this amplification is not what is driving cell death but rather something else that does not allow these cells to be maintained/progress”*,

We have picked up on this remark and made efforts to explore if these additional centrioles are either preventing pre B cells from differentiating into IgM⁺ naïve B cells. This, however, does not seem to be the case, as shown in Figure 3 (g-i), where we have compared the differentiation capacity of cells with high vs. low centriole count (see also Fig. S1d for validation). What became clear though is that during differentiation these structures are eliminated (see Figure 3h). As apoptosis is quite high in this assay, but physiologically not relevant (Figure 3d), we reasoned it may not be related to targeted removal of these cells, considering autophagic clearance or proteasomal degradation. Towards this end, we can show in Fig. 3j,k, that inhibition of autophagy by chloroquine does not prevent the disappearance of extra centrioles, pointing towards elimination in the proteasome. Combining MG132 with IL7 deprivation, however, was tricky, as, yes, cells with extra centrioles persisted after 24h (Fig. 3 l), but also more cells were found to accumulate in G2/M in the presence of MG132, where centriole amplification arises (Fig. 3m), rendering this result less interpretable. Please note that these experiments needed to be performed on a BCL2 transgenic background, as IL7-deprivation in combination with proteasome inhibition was not tolerated in wild type cells. We now present and discuss these findings on page 8.

Overall, we think our data is robust and the degree of novelty of our work is significant, meeting the expectation of the readership of Nature Communications. We sincerely hope that this referee will feel the same after reading our lines of argument and judging the new data provided.

On behalf of all authors, Andreas Villunger, PhD